# Bayesian Analysis of Intraday Stochastic Volatility Models of High-Frequency Stock Returns with Skew Heavy-Tailed Errors

**Makoto Nakakita [1],* and Teruo Nakatsuma [2]**

1   Centre for Finance, Technology and Economics at Keio (FinTEK), Keio University, Tokyo 108-8345, Japan
2   Faculty of Economics, Keio University, Tokyo 108-8345, Japan; nakatuma@econ.keio.ac.jp
*   Correspondence: nakakita@keio.jp

**Abstract:** Intraday high-frequency data of stock returns exhibit not only typical characteristics (e.g., volatility clustering and the leverage effect) but also a cyclical pattern of return volatility that is known as intraday seasonality. In this paper, we extend the stochastic volatility (SV) model for application with such intraday high-frequency data and develop an efficient Markov chain Monte Carlo (MCMC) sampling algorithm for Bayesian inference of the proposed model. Our modeling strategy is two-fold. First, we model the intraday seasonality of return volatility as a Bernstein polynomial and estimate it along with the stochastic volatility simultaneously. Second, we incorporate skewness and excess kurtosis of stock returns into the SV model by assuming that the error term follows a family of generalized hyperbolic distributions, including variance-gamma and Student's $t$ distributions. To improve efficiency of MCMC implementation, we apply an ancillarity-sufficiency interweaving strategy (ASIS) and generalized Gibbs sampling. As a demonstration of our new method, we estimate intraday SV models with 1 min return data of a stock price index (TOPIX) and conduct model selection among various specifications with the widely applicable information criterion (WAIC). The result shows that the SV model with the skew variance-gamma error is the best among the candidates.

**Keywords:** Bayesian inference; high-frequency financial time series; intraday seasonality; Markov chain Monte Carlo; stochastic volatility





## 1. Introduction

It is well documented that (a) probability distributions of stock returns are heavy-tailed (both tails of the probability density function go down to zero much slower than in the case of the normal distribution, and as a result, the kurtosis of the distribution exceeds 3), (b) they are often asymmetric around the mean (the skewness of the distribution is either positive or negative), (c) they exhibit volatility clustering (positive autocorrelation among the day-to-day variance of returns) and (d) the leverage effect (the current volatility and the previous return are negatively correlated so that downturns in the stock market tend to predate sharper spikes in the volatility). In the practice of financial risk management, it is imperative to develop a statistical model that can capture these characteristics of stock returns because they are thought to be related to steep drops and rebounds in stock prices during the periods of financial turmoil. Without factoring them into risk management, financial institutions might unintentionally take on a higher risk and as a result would be faced with grave consequences, which we already observed during the Global Financial Crisis.

As a time-series model with the aforementioned characteristics, a family of time-series models called the stochastic volatility (SV) model has been developed in the field of financial econometrics. The standard SV model is a simple state-space model in which the measurement equation is a mere distribution of stock returns with the time-varying variance (volatility) and the system equation is an AR(1) process of the latent log volatility. In the standard setting, both measurement and system errors are supposed to be Gaussian

and negatively correlated in order to incorporate the leverage effect into the model. The standard SV model can explain three stylized facts: heavy-tailed distribution, volatility clustering and the leverage effect, but it cannot make the distribution of stock returns asymmetric. Furthermore, although in theory the standard SV model incorporates the heavy-tail behavior of stock returns, many empirical studies demonstrated that it was insufficient to explain extreme fluctuations of stock prices that were caused by large shocks in financial markets.

Based on the plain-vanilla SV model, researchers have developed numerous variants that are designed to capture all aspects of stock returns sufficiently well. The SV model has been pioneered by Taylor (1982), and numerous studies related to the SV model have been conducted so far. The Markov chain Monte Carlo (MCMC) algorithms for SV models, which can be analyzed by numerical method, have been introduced by (Jacquier et al. 1994, 2004). Ghysels et al. (1996) also survey and develop statistical inferences of the SV model including a Bayesian approach. A direct way to introduce a more heavy-tailed distribution to the SV model is to assume that the error term of the measurement equation follows a distribution with much heavier tails than the normal distribution. The Student's $t$ distribution is a popular choice (Berg et al. 2004; Omori et al. 2007; Nakajima and Omori 2009; Nakajima 2012 among others). In the literature, the asymmetry in stock returns can be handled by assuming that the error term follows an asymmetric distribution (Nakajima and Omori 2012; Tsiotas 2012; Abanto-Valle et al. 2015 among others). In particular, the generalized hyperbolic (GH) distribution proposed by Barndorff-Nielsen (1977) has recently drawn increasing attention among researchers (e.g., Nakajima and Omori 2012), since it is regarded as a broad family of heavy-tailed distributions such as variance-gamma and Student's $t$, as well as their skewed variants such as skew variance-gamma and skew Student's $t$.

As an alternative to the SV model, the realized volatility (RV) model (e.g., Andersen and Bollerslev 1997, 1998) is often applied to evaluation of daily volatility. A naive RV estimator is defined as the sum of squared intraday returns. It converges to the daily integrated volatility as the time interval of returns becomes shorter. Due to this characteristic, RV is suitable for foreign exchange markets, which are open for 24 h a day continuously, though this may not be the case for stock markets. Most stock markets close at night, and some of them, including the Tokyo Stock Exchange, have lunch breaks when no transactions take place. It is well known that the naive RV estimator is biased for such stock markets. Nonetheless, since RV is a convenient tool for volatility estimation, researchers have developed various improved estimators of RV as well as robust estimators of its standard error. For example, Mykland and Zhang (2017) proposed a general nonparametric method called the observed asymptotic variance for assessing the standard error of RV.

Traditionally, empirical studies with the SV model as well as the RV model focused on daily volatility of asset returns. However, the availability of high-frequency tick data and the advent of high-frequency trading (HFT), which is a general term for algorithmic trading in full use of high-performance computing and high speed communication technology, has shifted the focus of research on volatility from closing-to-closing daily volatility to intraday volatility in a very short interval (e.g., 5 min or shorter). This shift paved the way for a new type of SV model. In addition to the traditional stylized facts on daily volatility, intraday volatility is known to exhibit a cyclical pattern during trading hours. On a typical trading day, the volatility tends to be high immediately after the market opens, but it gradually declines in the middle of trading hours. In the late trading hours, the volatility again becomes higher as it nears the closing time. This U-shaped trend in volatility is called intraday seasonality in the literature (see Chan et al. 1991 among others). Although it is crucial to take the intraday seasonality into consideration in estimation of any intraday volatility models, only a few studies (e.g., Stroud and Johannes 2014; Fičura and Witzany 2015a, 2015b) explicitly incorporate it into their volatility models.

In this paper, we propose to directly embed intraday seasonality into the SV model by approximating the U-shaped seasonality pattern with a linear combination of Bernstein

polynomials. In order to capture skewness and excess kurtosis in high-frequency stock returns, we employ two distributions (variance-gamma and Student's *t*) and their skewed variants (skew variance-gamma and skew Student's *t*) in the family of GH distributions as the distribution of stock returns in the SV model. The complicated SV models generally tend to be inefficient for analyzing in a primitive form. In order to solve the problem, numerous studies concerned with efficiency of the SV model have been developed. Omori and Watanabe (2008) introduce a sampling method with block unit for asymmetric SV models, which can sample disturbances from their conditional posterior distribution simultaneously. As another approach to optimize computation, a Sequential Monte Carlo (SMC) algorithm for Bayesian semi-parametric SV model was designed by Virbickaitė et al. (2019). The ancillarity-sufficiency interweaving strategy (ASIS) proposed by Yu and Meng (2011) is highly effective to improve MCMC sampling effeciency. We discuss ASIS in detail in Section 3. Needless to say, since the proposed SV model is intractably complicated, we develop an efficient Markov chain Monte Carlo (MCMC) sampling algorithm for full Bayesian estimation of all parameters and state variables (latent log volatilities in our case) in the model.

The rest of this paper is organized as follows. In Section 2, we introduce a reparameterized Gaussian SV model with leverage and intraday seasonality and derive an efficient MCMC sampling algorithm for its Bayesian estimation. In addition, we show the conditional posterior distributions and prepare for application of ASIS. In Section 3, we extend the Gaussian SV model to the case of variance gamma and Student's *t* error as well as their skewed variants. In Section 4, we report the estimation results of our proposed SV models with 1 min return data of TOPIX. Finally, conclusions are given in Section 5.

## 2. Stochastic Volatility Model with Intraday Seasonality

Consider the log difference of a stock price in a short interval (say, 1 or 5 min). We divide trading hours evenly into $T$ periods and normalize them so that the length of the trading hours is equal to 1; that is, the length of each period is $\frac{1}{T}$ and the time stamp of the $t$-th period is $\frac{t}{T}$ ($t = 1, \ldots, T$). Note that the market opens at time 0 and closes at time 1 in our setup. Let $y_t$ ($t = 1, \ldots, T$) denote the stock return in the $t$-th period (at time $\frac{t}{T}$ in the trading hours) and consider the following stochastic volatility (SV) model of $y_t$ with intraday seasonality:

$$\begin{cases} y_t = \exp(x_t'\beta + h_t)\epsilon_t, \\ h_{t+1} = \phi h_t + \eta_t, \end{cases} \quad \begin{bmatrix} \epsilon_t \\ \eta_t \end{bmatrix} \sim \text{Normal}\left( \begin{bmatrix} 0 \\ 0 \end{bmatrix}, \begin{bmatrix} 1 & \rho\tau \\ \rho\tau & \tau^2 \end{bmatrix} \right), \quad |\rho| < 1, \quad \tau > 0, \quad (1)$$

and

$$h_1 \sim \text{Normal}\left( 0, \frac{\tau^2}{1 - \phi^2} \right), \quad |\phi| < 1.$$

It is well known that the estimate of the correlation coefficient $\rho$ is negative in most stock markets. This negative correlation is often referred to as the leverage effect. Note that the stock volatility in the $t$-th period (the natural logarithm of the conditional standard deviation of $y_t$) is

$$\log \sqrt{\text{Var}[y_t | \mathcal{F}_{t-1}]} = x_t'\beta + h_t,$$

where $\mathcal{F}_{t-1}$ is the filtration that represents all available information at time $\frac{t-1}{T}$. Hence, the stock volatility in the SV model (1) is decomposed into two parts: a linear combination of covariates $x_t'\beta$ and the unobserved AR(1) process $h_t$. In this paper, we regard $x_t'\beta$ as the intraday seasonal component of the stock volatility, though it can be interpreted as any function of covariates $x_t$ in a different situation. On the other hand, $h_t$ is supposed to capture volatility clustering. We call $h_t$ the latent log volatility since it is unobservable.

Although the intraday seasonal component $x_t'\beta$ is likely to be a U-shaped function of time stamps (the stock volatility is higher right after the opening or near the closing, but it is lower in the middle of the trading hours), we have no information about the exact

functional form of the intraday seasonality. To make it in a flexible functional form for the intraday seasonality, we assume that $x_t'\beta$ is a Bernstein polynomial

$$x_t'\beta = \sum_{k=0}^{n} \beta_k x_{k,t} = \sum_{k=0}^{n} \beta_k b_{k,n}\left(\frac{t}{T}\right), \tag{2}$$

where $b_{k,n}(\cdot)$ is called a Bernstein basis polynomial of degree $n$:

$$b_{k,n}(v) = {}_nC_k v^k (1-v)^{n-k}, \quad k = 0, \dots, n, \quad v \in [0,1].$$

According to the Weierstrass approximation theorem, the Bernstein polynomial (2) can approximate any continuous function on $[0,1]$ as $n$ goes to infinity. In practice, however, the number of observations $T$ is finite. Thus, we need to choose a finite $n$ via a model selection procedure. We will discuss this issue in Section 4.

Although the parameterization of the SV model in (1) is widely applied in the literature, we propose an alternative parameterization that facilitates MCMC implementation in non-Gaussian SV models. By replacing the covariance matrix in (1) with

$$\begin{bmatrix} \mathrm{Var}[\epsilon_t] & \mathrm{Cov}[\eta_t, \epsilon_t] \\ \mathrm{Cov}[\epsilon_t, \eta_t] & \mathrm{Var}[\eta_t] \end{bmatrix} = \begin{bmatrix} 1 + \gamma^2\tau^2 & \gamma\tau^2 \\ \gamma\tau^2 & \tau^2 \end{bmatrix}, \quad \gamma \in \mathbb{R}, \tag{3}$$

we obtain an alternative formulation of the SV model:

$$\begin{cases} y_t = \exp(x_t'\beta + h_t)\epsilon_t, \\ h_{t+1} = \phi h_t + \eta_t, \end{cases} \begin{bmatrix} \epsilon_t \\ \eta_t \end{bmatrix} \sim \mathrm{Normal}\left(\begin{bmatrix} 0 \\ 0 \end{bmatrix}, \begin{bmatrix} 1 + \gamma^2\tau^2 & \gamma\tau^2 \\ \gamma\tau^2 & \tau^2 \end{bmatrix}\right). \tag{4}$$

Since in (4) the variance of $\epsilon_t$ is no longer equal to one, the interpretation of $\beta$ and $h_t$ in (4) is slightly different from the original one in (1). Nonetheless, the SV model (4) has essentially the same characteristics as (1). Since the correlation coefficient in (3) is

$$\mathrm{Corr}[\epsilon_t, \eta_t] = \frac{\gamma\tau}{\sqrt{1 + \gamma^2\tau^2}},$$

the sign of $\gamma$ always coincides with the correlation coefficient and the leverage effect exists if $\gamma < 0$. To distinguish $\gamma$ in (4) from the correlation parameter $\rho$ in (1), we call $\gamma$ the leverage parameter in this paper.

Note that the inverse of (3) is

$$\begin{bmatrix} \mathrm{Var}[\epsilon_t] & \mathrm{Cov}[\eta_t, \epsilon_t] \\ \mathrm{Cov}[\epsilon_t, \eta_t] & \mathrm{Var}[\eta_t] \end{bmatrix}^{-1} = \begin{bmatrix} 1 & -\gamma \\ -\gamma & \gamma^2 + \tau^{-2} \end{bmatrix} = \begin{bmatrix} 1 & 0 \\ -\gamma & \tau^{-1} \end{bmatrix}\begin{bmatrix} 1 & -\gamma \\ 0 & \tau^{-1} \end{bmatrix},$$

and the determinant of (3) is $\tau^2$. Using

$$\begin{bmatrix} \epsilon_t & \eta_t \end{bmatrix}\begin{bmatrix} 1 & -\gamma \\ -\gamma & \gamma^2 + \tau^{-2} \end{bmatrix}\begin{bmatrix} \epsilon_t \\ \eta_t \end{bmatrix} = \begin{bmatrix} \epsilon_t & \eta_t \end{bmatrix}\begin{bmatrix} 1 & 0 \\ -\gamma & \tau^{-1} \end{bmatrix}\begin{bmatrix} 1 & -\gamma \\ 0 & \tau^{-1} \end{bmatrix}\begin{bmatrix} \epsilon_t \\ \eta_t \end{bmatrix}$$

$$= (\epsilon_t - \gamma\eta_t)^2 + \frac{\eta_t^2}{\tau^2},$$

we can easily show that the SV model (4) is equivalent to

$$\begin{cases} y_t = \exp(x_t'\beta + h_t)(z_t + \gamma\eta_t), \\ h_{t+1} = \phi h_t + \eta_t, \end{cases} \tag{5}$$

where

$$z_t \sim \mathrm{Normal}(0,1), \ \eta_t \sim \mathrm{Normal}(0,\tau^2), \ z_t \perp \eta_t.$$

In the alternative formulation of the SV model (5), we can interpret $\eta_t$ as a common shock that affects both the stock return $y_t$ and the log volatility $h_{t+1}$ and $z_t$ as an idiosyncratic shock that affects $y_t$ only.

The likelihood for the SV model (5) given the observations $y_{1:T} = [y_1; \ldots; y_T]$, and the latent log volatility $h_{1:T+1} = [h_1; \ldots; h_{T+1}]$ is

$$p(y_{1:T}, h_{1:T+1}|\theta) = \underbrace{\prod_{t=1}^{T} p(y_t|h_t, h_{t+1}, \theta)}_{p(y_{1:T}|h_{1:T+1},\theta)} \cdot \underbrace{p(h_1|\theta) \prod_{t=1}^{T} p(h_{t+1}|h_t, \theta)}_{p(h_{1:T+1}|\theta)}, \qquad (6)$$

where

$$p(y_t|h_t, h_{t+1}, \theta) = \frac{1}{\sqrt{2\pi}} \exp\left[ -x_t'\beta - h_t - \frac{\{y_t \exp(-x_t'\beta - h_t) - \gamma(h_{t+1} - \phi h_t)\}^2}{2} \right], \quad (7)$$

$$p(h_{t+1}|h_t, \theta) = \frac{1}{\sqrt{2\pi\tau^2}} \exp\left[ -\frac{(h_{t+1} - \phi h_t)^2}{2\tau^2} \right], \quad t = 1, \ldots, T,$$

$$p(h_1|\theta) = \sqrt{\frac{1 - \phi^2}{2\pi\tau^2}} \exp\left[ -\frac{(1 - \phi^2)h_1^2}{2\tau^2} \right],$$

and $\theta = (\beta, \gamma, \tau^2, \phi)$. Since $h_t$ follows a stationary AR(1) process, the joint probability distribution of $h_{1:T+1}$ is Normal$(0, \tau^2 V^{-1})$, where

$$V = \begin{bmatrix} 1 & -\phi & & & & & \\ -\phi & 1+\phi^2 & -\phi & & & & \\ & -\phi & 1+\phi^2 & -\phi & & & \\ & & \ddots & \ddots & \ddots & & \\ & & & -\phi & 1+\phi^2 & -\phi & \\ & & & & -\phi & 1+\phi^2 & -\phi \\ & & & & & -\phi & 1 \end{bmatrix}, \qquad (8)$$

is a tridiagonal matrix, and it is positive definite as long as $|\phi| < 1$. Thus, the joint p.d.f. of $h_{1:T+1}$ is

$$p(h_{1:T+1}|\theta) = (2\pi\tau^2)^{-\frac{T+1}{2}} |V|^{\frac{1}{2}} \exp\left[ -\frac{1}{2\tau^2} h_{1:T+1}' V h_{1:T+1} \right], \quad |V| = 1 - \phi^2. \qquad (9)$$

The prior distributions for $(\beta, \gamma, \tau^2, \phi)$ in our study are

$$\beta \sim \text{Normal}(\bar{\mu}_\beta, \bar{\Omega}_\beta^{-1}), \quad \gamma \sim \text{Normal}(\bar{\mu}_\gamma, \bar{\omega}_\gamma^{-1}),$$

$$\tau^2 \sim \text{Inv. Gamma}(a_\tau, b_\tau), \quad \frac{\phi + 1}{2} \sim \text{Beta}(a_\phi, b_\phi). \qquad (10)$$

Then the joint posterior density of $(h_{1:T+1}, \theta)$ for the SV model (5) is

$$p(h_{1:T+1}, \theta|y_{1:T}) \propto \prod_{t=1}^{T} p(y_t|h_t, h_{t+1}, \theta) \cdot p(h_{1:T+1}|\theta) \cdot p(\theta), \qquad (11)$$

where $p(\theta)$ is the prior density of the parameters in (10).

Since analytical evaluation of the joint posterior distribution (11) is impractical, we apply an MCMC method to generate a random sample $\{(h_{1:T+1}^{(r)}, \beta^{(r)}, \gamma^{(r)}, \tau^{2(r)}, \phi^{(r)})\}_{r=1}^{R}$ from the joint posterior distribution (11) and numerically evaluate the posterior statistics necessary for Bayesian inference with Monte Carlo integration. The outline of the standard MCMC sampling scheme for the posterior distribution (11) is given as follows:

---

***Outline of the MCMC sampling for the SV model***

**Step 0:** Initialize $(h_{1:T+1}^{(0)}, \beta^{(0)}, \gamma^{(0)}, \tau^{2(0)}, \phi^{(0)})$ and set the counter $r = 0$.

**Step 1:** Generate $(h_{1:T+1}^{(r+1)}, \beta^{(r+1)}, \gamma^{(r+1)}, \tau^{2(r+1)}, \phi^{(r+1)})$ with the following scheme:

    **Step 1-1:** Generate $h_{1:T+1}^{(r+1)}$ from $p(h_{1:T+1}|\beta^{(r)}, \gamma^{(r)}, \tau^{2(r)}, \phi^{(r)}, y_{1:T})$.

    **Step 1-2:** Generate $\beta^{(r+1)}$ from $p(\beta|h_{1:T+1}^{(r+1)}, \gamma^{(r)}, \tau^{2(r)}, \phi^{(r)}, y_{1:T})$.

    **Step 1-3:** Generate $\gamma^{(r+1)}$ from $p(\gamma|h_{1:T+1}^{(r+1)}, \beta^{(r+1)}, \tau^{2(r)}, \phi^{(r)}, y_{1:T})$

    **Step 1-4:** Generate $\tau^{2(r+1)}$ from $p(\tau|h_{1:T+1}^{(r+1)}, \beta^{(r+1)}, \gamma^{(r+1)}, \phi^{(r)}, y_{1:T})$

    **Step 1-5:** Generate $\phi^{(r+1)}$ from $p(\phi|h_{1:T+1}^{(r+1)}, \beta^{(r+1)}, \gamma^{(r+1)}, \tau^{2(r+1)}, y_{1:T})$.

**Step 2:** Let $r = r + 1$ and go to **Step 1** until the burn-in iterations are completed.

**Step 3:** Reset the counter $r = 0$ and repeat **Step 1–2** $R$ times in order to obtain the Monte Carlo sample $\{(h_{1:T+1}^{(r)}, \beta^{(r)}, \gamma^{(r)}, \tau^{2(r)}, \phi^{(r)})\}_{r=1}^{R}$.

---

Although the above MCMC sampling scheme is ubiquitous in the literature of the SV model, the generated Monte Carlo sample $\{(h_{1:T+1}^{(r)}, \beta^{(r)}, \gamma^{(r)}, \tau^{2(r)}, \phi^{(r)})\}_{r=1}^{R}$ tends to exhibit strongly positive autocorrelation. To improve efficiency of MCMC implementation, Yu and Meng (2011) proposed an ancillarity-sufficiency interweaving strategy (ASIS). In the literature of the SV model, Kastner and Frühwirth-Schnatter (2014) applied ASIS to the SV model of daily US-dollar/Euro exchange rate data with the Gaussian error. Their SV model did not include either intraday seasonality or the leverage effect since they applied it to daily exchange rate data that exhibited no leverage effect in most cases. We extend the algorithm developed by Kastner and Frühwirth-Schnatter (2014) to facilitate the converge of the sample path in the SV model (5). The basic principle of ASIS is to construct MCMC sampling schemes for two different but equivalent parameterizations of a model with missing/latent variables ($h_{1:T+1}$ in our case) and generate the parameters alternately with each of them.

According to Kastner and Frühwirth-Schnatter (2014), the SV model (5) is in a non-centered parameterization (NCP). On the other hand, we may transform $h_t$ as

$$\tilde{h}_t = x_t'\beta + h_t, \tag{12}$$

and rearrange the SV model (5) as

$$\begin{cases} y_t = \exp(\tilde{h}_t)(z_t + \gamma\eta_t), \\ \tilde{h}_{t+1} - x_{t+1}'\beta = \phi(\tilde{h}_t - x_t'\beta) + \eta_t. \end{cases} \tag{13}$$

The above SV model (13) is in a centered parameterization (CP).

The posterior distribution in the CP form (13) is equivalent to the one in the NCP form (5) in the sense that they give us the same posterior distribution of $\theta$. Let us verify this claim. The likelihood for the SV model (13) given the observations $y_{1:T}$ and the latent log volatility $\tilde{h}_{1:T+1} = [\tilde{h}_1; \ldots; \tilde{h}_{T+1}]$ is

$$p(y_{1:T}, \tilde{h}_{1:T+1}|\theta) = \underbrace{\prod_{t=1}^{T} p(y_t|\tilde{h}_t, \tilde{h}_{t+1}, \theta)}_{p(y_{1:T}|\tilde{h}_{1:T+1}, \theta)} \cdot \underbrace{p(\tilde{h}_1|\theta)\prod_{t=1}^{T} p(\tilde{h}_{t+1}|\tilde{h}_t, \theta)}_{p(\tilde{h}_{1:T+1}|\theta)}, \tag{14}$$

where

$$p(y_t|\tilde{h}_t, \tilde{h}_{t+1}, \theta) = \frac{1}{\sqrt{2\pi}} \exp\left[-\tilde{h}_t - \frac{\left\{y_t e^{-\tilde{h}_t} - \gamma((\tilde{h}_{t+1} - x'_{t+1}\beta) - \phi(\tilde{h}_t - x'_t\beta))\right\}^2}{2}\right], \tag{15}$$

$$p(\tilde{h}_{t+1}|\tilde{h}_t, \theta) = \frac{1}{\sqrt{2\pi\tau^2}} \exp\left[-\frac{\{(\tilde{h}_{t+1} - x'_{t+1}\beta) - \phi(\tilde{h}_t - x'_t\beta)\}^2}{2\tau^2}\right], \quad t = 1, \ldots, T,$$

$$p(\tilde{h}_1|\theta) = \sqrt{\frac{1 - \phi^2}{2\pi\tau^2}} \exp\left[-\frac{(1 - \phi^2)(\tilde{h}_1 - x'_1\beta)^2}{2\tau^2}\right].$$

Note that the joint p.d.f. of $\tilde{h}_{1:T+1}$ is

$$p(\tilde{h}_{1:T+1}|\theta) = (2\pi\tau^2)^{-\frac{T+1}{2}} |V|^{\frac{1}{2}} \exp\left[-\frac{1}{2\tau^2}(\tilde{h}_{1:T+1} - X\beta)'V(\tilde{h}_{1:T+1} - X\beta)\right], \tag{16}$$

where $X = [x'_1; \ldots; x'_{T+1}]$. With the prior of $\theta$ in (10), the joint posterior density of $(\tilde{h}_{1:T+1}, \theta)$ for the SV model (13) is obtained as

$$p(\tilde{h}_{1:T+1}, \theta|y_{1:T}) \propto \prod_{t=1}^{T} p(y_t|\tilde{h}_t, \tilde{h}_{t+1}, \theta) \cdot p(\tilde{h}_{1:T+1}|\theta) \cdot p(\theta). \tag{17}$$

Note that $\theta$ is unchanged between the NCP form (11) and the CP form (17). Although the latent variables are transformed with (12), the "marginal" posterior p.d.f. of $\theta$ is unchanged, because

$$\int p(\tilde{h}_{1:T+1}, \theta|y_{1:T+1}) d\tilde{h}_{1:T+1} = \int p(h_{1:T+1}, \theta|y_{1:T+1})|J| dh_{1:T+1},$$

where the Jacobian $|J| = 1$.

With this fact in mind, we can incorporate ASIS into the MCMC sampling scheme by replacing **Step 1** with

---

*NCP-based ASIS step*

**Step 1:** Generate $(h_{1:T+1}^{(r+0.5)}, \beta^{(r+0.5)}, \gamma^{(r+0.5)}, \tau^{2(r+0.5)}, \phi^{(r+0.5)})$ with the sampling scheme based on the NCP form (5) and compute

$$\tilde{h}_t^{(r+0.5)} = h_t^{(r+0.5)} + x'_t\beta^{(r+0.5)}, \quad t = 1, \ldots, T+1.$$

**Step 1.5:** Generate $(\beta^{(r+1)}, \gamma^{(r+1)}, \tau^{2(r+1)}, \phi^{(r+1)})$ with the sampling scheme based on the CP form (13) and compute

$$h_t^{(r+1)} = \tilde{h}_t^{(r+0.5)} - x'_t\beta^{(r+1)}, \quad t = 1, \ldots, T+1.$$

---

Note that we generate a new latent log volatility $h_{1:T+1}$ from its conditional posterior distribution in the NCP form (11) only once at the beginning of **Step 1**. This is the reason we call it the NCP-based ASIS step. After this update, we merely shift the location of $h_{1:T+1}$ by $x'_t\beta^{(r+0.5)}$ (**Step 1**) or by $-x'_t\beta^{(r+1)}$ (**Step 1.5**). In ASIS, these shifts are applied with probability 1 even if all elements in $h_{1:T+1}$ are not updated at the beginning of **Step 1**, which is highly probable in practice because we need to use the MH algorithm to generate $h_{1:T+1}$. Although we also utilize the MH algorithm to generate $\beta$, as explained later, the acceptance rate of $\beta$ in the MH step is much higher than that of $h_{1:T+1}$ in our experience. Thus, we expect that both $x'_t\beta^{(r+0.5)}$ and $-x'_t\beta^{(r+1)}$ will be updated more often than $h_{1:T+1}$ itself. As a result, the above ASIS step may improve mixing of the sample sequence of $h_{1:T+1}$. Conversely, we may apply the following CP-based ASIS step:

---

***CP-based ASIS step***

**Step 1:** Generate $(\tilde{h}_{1:T+1}^{(r+0.5)}, \beta^{(r+0.5)}, \gamma^{(r+0.5)}, \tau^{2(r+0.5)}, \phi^{(r+0.5)})$ with the sampling scheme based on the CP form (13) and compute

$$h_t^{(r+0.5)} = \tilde{h}_t^{(r+0.5)} - x_t'\beta^{(r+0.5)}, \quad t = 1, \ldots, T+1.$$

**Step 1.5:** Generate $(\beta^{(r+1)}, \gamma^{(r+1)}, \tau^{2(r+1)}, \phi^{(r+1)})$ with the sampling scheme based on the NCP form (5) and compute

$$\tilde{h}_t^{(r+1)} = h_t^{(r+0.5)} + x_t'\beta^{(r+1)}, \quad t = 1, \ldots, T+1.$$

---

In the CP-based ASIS step, we generate $\tilde{h}_{1:T+1}$ from its conditional posterior distribution in the CP form (17) once. The rest is the same as in the NCP-based ASIS step except that the order of sampling is reversed.

In the NCP form, the conditional posterior distributions for $(\beta, \gamma, \tau^2, \phi)$ are

$$\beta \sim \text{Normal}\big(\mu_\beta(\beta^*), \Sigma_\beta(\beta^*)\big), \tag{18}$$

where

$$\Sigma_\beta(\beta^*) = \big(Q(\beta^*) + \bar{\Omega}_\beta\big)^{-1}, \quad \mu_\beta(\beta^*) = \Sigma_\beta(\beta^*)\big(g(\beta^*) + Q(\beta^*)\beta^* + \bar{\Omega}_\beta\bar{\mu}_\beta\big).$$

$$\gamma | h_{1:T+1}, \theta_{-\gamma}, y_{1:T} \sim \text{Normal}\left(\frac{\sum_{t=1}^T \eta_t \epsilon_t + \bar{\omega}_\gamma \bar{\mu}_\gamma}{\sum_{t=1}^T \eta_t^2 + \bar{\omega}_\gamma}, \frac{1}{\sum_{t=1}^T \eta_t^2 + \bar{\omega}_\gamma}\right). \tag{19}$$

$$\tau^2 | h_{1:T+1}, \theta_{-\tau^2}, y_{1:T} \sim \text{Inv. Gamma}\left(\frac{T+1}{2} + a_\tau, \frac{1}{2}h_{1:T+1}'V h_{1:T+1} + b_\tau\right). \tag{20}$$

$$\phi \sim \text{Normal}\left(\frac{\sum_{t=1}^T h_{t+1}h_t}{\sum_{t=2}^T h_t^2}, \frac{\tau^2}{\sum_{t=2}^T h_t^2} \,\bigg|\, -1 < \phi < 1\right). \tag{21}$$

In the CP form, the conditional posterior distributions for $(\beta, \gamma, \tau^2, \phi)$ are

$$\beta \sim \text{Normal}\big(\tilde{\mu}_\beta, \tilde{\Sigma}_\beta\big), \tag{22}$$

where

$$\tilde{\Sigma}_\beta = \left(\tilde{X}'\tilde{X} + \frac{1}{\tau^2}X'VX + \bar{\Omega}_\beta\right)^{-1},$$

$$\tilde{\mu}_\beta = \tilde{\Sigma}_\beta\left(\tilde{X}'\tilde{y} + \frac{1}{\tau^2}X'V\tilde{h}_{1:T+1} + \bar{\Omega}_\beta\bar{\mu}_\beta\right).$$

$$\gamma | \tilde{h}_{1:T+1}, \theta_{-\gamma}, y_{1:T} \sim \text{Normal}\left(\frac{\sum_{t=1}^T \tilde{\eta}_t \tilde{\epsilon}_t + \bar{\omega}_\gamma \bar{\mu}_\gamma}{\sum_{t=1}^T \tilde{\eta}_t^2 + \bar{\omega}_\gamma}, \frac{1}{\sum_{t=1}^T \tilde{\eta}_t^2 + \bar{\omega}_\gamma}\right). \tag{23}$$

$$\tau^2 | \tilde{h}_{1:T+1}, \theta_{-\tau^2}, y_{1:T} \sim \text{Inv. Gamma}\left(\frac{T+1}{2}a_\tau, \frac{1}{2}(\tilde{h}_{1:T+1} - X\beta)'V(\tilde{h}_{1:T+1} - X\beta) + b_\tau\right). \tag{24}$$

$$\phi \sim \text{Normal}\left(\frac{\sum_{t=1}^T (\tilde{h}_{t+1} - x_{t+1}'\beta)(\tilde{h}_t - x_t'\beta)}{\sum_{t=2}^T (\tilde{h}_t - x_t'\beta)^2}, \frac{\tau^2}{\sum_{t=2}^T (\tilde{h}_t - x_t'\beta)^2} \,\bigg|\, -1 < \phi < 1\right). \tag{25}$$

Derivations of the conditional posterior distriburions are shown in Appendix A.

## 3. Extension: Skew Heavy-Tailed Distributions

*3.1. Mean-Variance Mixture of the Normal Distribution*

It is a well-known stylized fact that probability distributions of stock returns are almost definitely heavy-tailed (the probability density goes down to zero much slower than the normal distribution) and often have non-zero skewness (they are not symmetric around the mean). Although introduction of stochastic volatility and leverage makes the distribution of $y_t$ skew and heavy-tailed, it may not be sufficient to capture those characteristics of real data. For this reason, instead of the normal distribution, we introduce a skew heavy-tailed distribution to the SV model.

In our study, we suppose that $z_t$ in (5) is expressed as a mean-variance mixture of the standard normal distribution:

$$z_t = \alpha \delta_t + \sqrt{\delta_t} u_t, \quad u_t \sim \text{Normal}(0, 1), \quad \delta_t \sim \text{GIG}(\lambda, \psi, \xi), \tag{26}$$

where $\text{GIG}(\lambda, \psi, \xi)$ stands for the generalized inverse Gaussian distribution with the probability density:

$$p(\delta_t) = \frac{(\psi/\xi)^{\lambda/2}}{2K_\lambda(\sqrt{\psi\xi})} \delta_t^{\lambda-1} \exp\left[-\frac{1}{2}\left(\psi\delta_t + \frac{\xi}{\delta_t}\right)\right], \tag{27}$$

where

$$\lambda \in \mathbb{R}, \quad (\psi, \xi) \in \begin{cases} \{(\psi, \xi) : \psi > 0, \xi \geq 0\} & \text{if } \lambda > 0, \\ \{(\psi, \xi) : \psi > 0, \xi > 0\} & \text{if } \lambda = 0, \\ \{(\psi, \xi) : \psi \geq 0, \xi > 0\} & \text{if } \lambda < 0, \end{cases}$$

and $K_\lambda(\cdot)$ is the modified Bessel function of the second kind. The family of generalized inverse Gaussian distributions includes

- exponential distribution ($\lambda = 1, \xi = 0$),
- gamma distribution ($\lambda > 0, \xi = 0$),
- inverse gamma distribution ($\lambda < 0, \psi = 0$),
- inverse Gaussian distribution ($\lambda = -\frac{1}{2}$)

Under the assumption (26), the distribution of $z_t$ belongs to the family of generalized hyperbolic distributions proposed by Barndorff-Nielsen (1977), which includes many well-known skew heavy-tailed distributions such as

- skew variance gamma (VG) distribution($\lambda = \frac{\nu}{2}, \psi = \nu, \xi = 0$),
- skew $t$ distribution ($\lambda = -\frac{\nu}{2}, \psi = 0, \xi = \nu$),

where $\nu > 0$. In general, the skew VG distribution is a mean-variance mixture of the standard normal distribution with $\text{GIG}(\lambda, \psi, 0)$. To make the estimation easier, we set $\lambda = \frac{\nu}{2}$ and $\psi = \nu$ so that the skew VG distribution has only two free parameters $(\alpha, \nu)$. Thus, we have two additional parameters $(\alpha, \nu)$ in the SV model. Since $\alpha$ determines whether the distribution of $y_t$ is symmetric or not while $\nu$ determines how heavy-tailed the distribution is, we call $\alpha$ the asymmetry parameter and $\nu$ the tail parameter, respectively. In our study, we use the above three skew heavy-tailed distributions as alternatives to the normal distribution. To distinguish each model specification, we use the following abbreviations:

| | |
|---:|:---|
| SV-N: | stochastic volatility model with the normal error, |
| SV-G: | stochastic volatility model with the VG error, |
| SV-SG: | stochastic volatility model with the skew VG error, |
| SV-T: | stochastic volatility model with the Student-$t$ error, |
| SV-ST: | stochastic volatility model with the skew $t$ error. |

In this setup, the SV model with heavy-tailed error is formulated as

$$
\begin{cases} y_t = \exp(x_t'\beta + h_t)\epsilon_t, \\ h_{t+1} = \phi h_t + \eta_t, \end{cases} \quad \begin{bmatrix} \epsilon_t \\ \eta_t \end{bmatrix} \Big| \delta_t \sim \text{Normal}\left( \begin{bmatrix} \alpha\delta_t \\ 0 \end{bmatrix}, \begin{bmatrix} \delta_t + \gamma^2\tau^2 & \gamma\tau^2 \\ \gamma\tau^2 & \tau^2 \end{bmatrix} \right). \tag{28}
$$

It is straightforward to show that the conditional probability density of $y_t$ given $(h_t, h_{t+1})$ is given by

$$
p(y_t|h_t, h_{t+1}, \theta) = \int_0^\infty p(y_t|h_t, h_{t+1}, \delta_t, \theta) p(\delta_t|\nu)\delta_t, \tag{29}
$$

where $\theta = (\beta, \gamma, \tau^2, \phi, \alpha, \nu)$,

$$
\begin{aligned}
& p(y_t|h_t, h_{t+1}, \delta_t, \theta) \\
& = \frac{1}{\sqrt{2\pi\delta_t}} \exp\left[ -x_t'\beta - h_t - \frac{\{y_t \exp(-x_t'\beta - h_t) - \alpha\delta_t - \gamma(h_{t+1} - \phi h_t)\}^2}{2\delta_t} \right],
\end{aligned} \tag{30}
$$

and

$$
p(\delta_t|\nu) = \begin{cases} \dfrac{(\nu/2)^{\nu/2}}{\Gamma(\nu/2)} \delta_t^{\frac{\nu}{2}-1} \exp\left( -\dfrac{\nu}{2}\delta_t \right) & \text{(SV-SG)}, \\[4mm] \dfrac{(\nu/2)^{\nu/2}}{\Gamma(\nu/2)} \delta_t^{-\frac{\nu}{2}-1} \exp\left( -\dfrac{\nu}{2\delta_t} \right) & \text{(SV-ST)}. \end{cases} \tag{31}
$$

Since it is impractical to evaluate the multiple integral in (29), we generate $\delta_{1:T} = (\delta_1, \ldots, \delta_T)$ along with $h_{1:T+1}$ and $\theta$ form their joint posterior distribution. In this setup, the likelihood used in the posterior simulation is

$$
\begin{aligned}
p(y_{1:T}, h_{1:T+1}, \delta_{1:T}|\theta) &= p(y_{1:T}|h_{1:T+1}, \delta_{1:T}, \theta) p(h_{1:T+1}|\theta) \\
&= \prod_{t=1}^{T} p(y_t|h_t, h_{t+1}, \theta) \cdot p(h_{1:T+1}|\theta).
\end{aligned} \tag{32}
$$

We suppose that the prior distributions for $\alpha$ and $\nu$ are

$$
\alpha \sim \text{Normal}(\bar{\mu}_\alpha, \bar{\omega}_\alpha^{-1}), \quad \nu \sim \text{Gamma}(a_\nu, b_\nu). \tag{33}
$$

As for the other parameters, we keep the same ones as in (10).

*3.2. Conditional Posterior Distributions*

3.2.1. Latent Log Volatility $h_{1:T+1}$

Our sampling scheme for $h_{1:T+1}$ is basically the same as before. We first approximate the log likelihood with the second-order Taylor expansion around the mode and construct a proposal distribution of $h_{1:T+1}$ with the approximated log likelihood. Then, we apply a multi-move MH sampler for generating $h_{1:T+1}$ from the conditional posterior distribution. The sole differences are the functional form of $g(h_{1:T+1})$ and $Q(h_{1:T+1})$.

$$
\begin{aligned}
g_t(h_{1:T+1}) = & \left\{ -1 + \frac{1}{\delta_t}(\epsilon_t - \alpha\delta_t - \gamma\eta_t)(\epsilon_t - \gamma\phi) \right\} \mathbf{1}(t \leqq T) \\
& + \frac{\gamma}{\delta_{t-1}}(\epsilon_{t-1} - \alpha\delta_{t-1} - \gamma\eta_{t-1})\mathbf{1}(t \geqq 2), \quad (t = 1, \ldots, T+1),
\end{aligned} \tag{34}
$$

where $\mathbf{1}(\cdot)$ is the indicator function. Each diagonal element of $Q(h_{1:T+1})$ is

$$
\begin{aligned}
q_{t,t}(h_{1:T+1}) = & \frac{1}{\delta_t}\left\{ \epsilon_t(\epsilon_t - \alpha\delta_t - \gamma\eta_t) + (\epsilon_t - \gamma\phi)^2 \right\} \mathbf{1}(t \leqq T) \\
& + \frac{\gamma^2}{\delta_{t-1}}\mathbf{1}(t \geqq 2), \quad (t = 1, \ldots, T+1),
\end{aligned} \tag{35}
$$

and the off-diagonal element is

$$q_{t,t+1}(h_{1:T+1}) = \frac{\gamma}{\delta_t}(\epsilon_t - \gamma\phi), \quad (t = 1, \dots, T). \tag{36}$$

For the NCP form, we use $\epsilon_t$ and $\eta_t$ in (A3). For the CP form, we replace them with $\tilde{\epsilon}_t$ and $\tilde{\eta}_t$ in (A23).

### 3.2.2. Regression Coefficients $\beta$

The sampling scheme for $\beta$ is the same as before. For the NCP form, $g(\beta)$ and $Q(\beta)$ are given by

$$g(\beta) = \sum_{t=1}^{T} \left( \frac{\epsilon_t}{\delta_t}(\epsilon_t - \alpha\delta_t - \gamma\eta_t) - 1 \right) x_t, \tag{37}$$

$$Q(\beta) = \sum_{t=1}^{T} \frac{\epsilon_t}{\delta_t}(2\epsilon_t - \alpha\delta_t - \gamma\eta_t)x_t x_t', \tag{38}$$

respectively. For the CP form, the conditional posterior distribution of $\beta$ are given by

$$\beta \sim \text{Normal}(\tilde{\mu}_\beta, \tilde{\Sigma}_\beta), \tag{39}$$

where

$$\tilde{\Sigma}_\beta = \left( \tilde{X}' D^{-1} \tilde{X} + \frac{1}{\tau^2} X' V X + \bar{\Omega}_\beta \right)^{-1},$$

$$\tilde{\mu}_\beta = \tilde{\Sigma}_\beta \left( \tilde{X}' D^{-1} \tilde{y} + \frac{1}{\tau^2} X' V \tilde{h}_{1:T+1} + \bar{\Omega}_\beta \bar{\mu}_\beta \right),$$

$$\tilde{y}_t = \tilde{\epsilon}_t - \alpha\delta_t - \gamma(\tilde{h}_{t+1} - \phi\tilde{h}_t), \quad \tilde{y} = [\tilde{y}_1; \dots; \tilde{y}_T], \quad D = \text{diag}\{\delta_1, \dots, \delta_T\}.$$

### 3.2.3. Leverage Parameter $\gamma$

Their conditional posterior distribution of $\gamma$ is given by

$$\gamma | h_{1:T+1}, \delta_{1:T}, \theta_{-\gamma}, y_{1:T} \sim \text{Normal}\left( \frac{\sum_{t=1}^{T} \eta_t(\epsilon_t/\delta_t - \alpha) + \bar{\omega}_\gamma \bar{\mu}_\gamma}{\sum_{t=1}^{T} \eta_t^2/\delta_t + \bar{\omega}_\gamma}, \frac{1}{\sum_{t=1}^{T} \eta_t^2/\delta_t + \bar{\omega}_\gamma} \right). \tag{40}$$

For the NCP form, we use $\epsilon_t$ and $\eta_t$ in (A3). For the CP form, we replace them with $\tilde{\epsilon}_t$ and $\tilde{\eta}_t$ in (A23).

### 3.2.4. Random Scale $\delta_{1:T}$

Using the Bayes theorem, we obtain the conditional posterior distribution of $\delta_t$ as

$$\delta_t | h_{1:T+1}, \theta, y_{1:T} \sim \text{GIG}(\lambda_t, \psi_t, \xi_t), \quad t = 1, \dots, T, \tag{41}$$

where

$$(\lambda_t, \psi_t, \xi_t) = \begin{cases} \left( \dfrac{\nu-1}{2}, \alpha^2 + \nu, (\epsilon_t - \gamma\eta_t)^2 \right), & \text{(SV-SG)}, \\ \left( -\dfrac{\nu+1}{2}, \alpha^2, (\epsilon_t - \gamma\eta_t)^2 + \nu \right), & \text{(SV-ST)}. \end{cases}$$

For the NCP form, we use $\epsilon_t$ and $\eta_t$ in (A3). For the CP form, we replace them with $\tilde{\epsilon}_t$ and $\tilde{\eta}_t$ in (A23).

To improve the performance of the MCMC algorithm, we apply a generalized Gibbs sampler by Liu and Sabatti (2000) to $\{\delta_t\}_{t=1}^{T}$ after we generate them from the conditional

posterior distribution (41). This is rather simple. All we need to do is to multiply each of $\{\delta_t\}_{t=1}^{T}$ by a random number $c$ that is generated from

$$
c \sim \begin{cases} \text{GIG}\left( \dfrac{(\nu-1)T}{2}, (\alpha^2+\nu)\sum_{t=1}^{T}\delta_t, \sum_{t=1}^{T}\dfrac{(\epsilon_t-\gamma\eta_t)^2}{\delta_t} \right), & \text{(SV-SG)}, \\[3mm] \text{GIG}\left( -\dfrac{(\nu+1)T}{2}, \alpha^2\sum_{t=1}^{T}\delta_t, \sum_{t=1}^{T}\dfrac{(\epsilon_t-\gamma\eta_t)^2+\nu}{\delta_t} \right), & \text{(SV-ST)}. \end{cases} \tag{42}
$$

### 3.2.5. Asymmetry Parameter $\alpha$

Using the Bayes theorem, we obtain the conditional posterior distribution of $\alpha$ as

$$
\begin{aligned} & \alpha | h_{1:T+1}, \delta_{1:T}, \theta_{-\alpha}, y_{1:T} \\ & \sim \text{Normal}\left( \frac{\sum_{t=1}^{T}(\epsilon_t-\gamma\eta_t)+\bar{\omega}_\alpha\bar{\mu}_\alpha}{\sum_{t=1}^{T}\delta_t+\bar{\omega}_\alpha}, \frac{1}{\sum_{t=1}^{T}\delta_t+\bar{\omega}_\alpha} \right). \end{aligned} \tag{43}
$$

For the NCP form, we use $\epsilon_t$ and $\eta_t$ in (A3). For the CP form, we replace them with $\tilde{\epsilon}_t$ and $\tilde{\eta}_t$ in (A23).

### 3.2.6. Tail Parameter $\nu$

The explicit form of the conditional posterior density of $\nu$ is not available. Therefore, we apply the MH algorithm for generating $\nu$. Note that the gamma density for SV-SG in (31) is identical to the inverse gamma density for SV-ST in (31) as a function of $\nu$ if we exchange $\delta_t$ with $\delta_t^{-1}$. Since we use the same gamma prior for $\nu$ in either case, the resultant conditional posterior density should be the same in both SV-SG and SV-ST. Therefore, it suffices to derive the MH algorithm for SV-ST.

The sampling strategy for $\nu$ is basically the same as for $\beta$, which was originally proposed by Watanabe (2001). We first consider the second-order Taylor expansion of the log conditional posterior density of $\nu$:

$$
f(\nu) = \sum_{t=1}^{T}\log p(\delta_t|\nu) + \log p(\nu) + \text{constant} \tag{44}
$$

$$
= \frac{\nu T}{2}\log\frac{\nu}{2} - T\log\Gamma\left(\frac{\nu}{2}\right) - \nu\left\{ \frac{1}{2}\sum_{t=1}^{T}\left(\log\delta_t + \frac{1}{\delta_t}\right) + b_\nu \right\} + (a_\nu-1)\log\nu \tag{45}
$$

$$
+ \text{constant},
$$

with respect to $\nu$ in the neighborhood of $\nu^* > 0$, i.e.,

$$
f(\nu) \approx f(\nu^*) + g(\nu^*)(\nu-\nu^*) - \frac{1}{2}q(\nu^*)(\nu-\nu^*)^2, \tag{46}
$$

where

$$
\begin{aligned} g(\nu^*) &\equiv \nabla_\nu f(\nu^*) \\ &= \frac{T}{2} + \frac{T}{2}\log\frac{\nu^*}{2} - \frac{T}{2}\psi^{(0)}\left(\frac{\nu^*}{2}\right) - \frac{1}{2}\sum_{t=1}^{T}\left(\log\delta_t + \frac{1}{\delta_t}\right) - b_\nu + \frac{a_\nu-1}{\nu^*}, \\ q(\nu^*) &\equiv -\nabla_\nu^2 f(\nu^*) \\ &= -\frac{T}{2\nu^*} + \frac{T}{4}\psi^{(1)}\left(\frac{\nu^*}{2}\right) + \frac{a_\nu-1}{\nu^{*2}}, \end{aligned}
$$

and $\psi^{(s)}$ is the polygamma function of order $s$. Note that $q(\nu^*) > 0$ if $T + 2a_\nu > 2$. See Theorem 1 in Watanabe (2001) for the proof. By applying the completing-the-square technique to (46), we obtain the proposal distribution for the MH algorithm:

$$\nu \sim \text{Normal}\Big(\mu_\nu(\nu^*), \sigma_\nu^2(\nu^*)\Big), \qquad (47)$$

where

$$\sigma_\nu^2(\nu^*) = \frac{1}{q(\nu^*)}, \quad \mu_\nu(\nu^*) = \nu^* + \frac{g(\nu^*)}{q(\nu^*)}.$$

If we use the mode of $f(\nu)$ as $\nu^*$, $g(\nu^*) = 0$ always holds due to the global concavity of $f(\nu)$. Thus, $\mu_\nu(\nu^*)$ is effectively identical to $\nu^*$.

## 4. Empirical Study

As an application of our proposed models to real data, we analyze high-frequency data of the Tokyo Stock Price Index (TOPIX), a market-cap-weighted stock index based on all domestic common stocks listed in the Tokyo Stock Exchange (TSE) First Section, which is provided by Nikkei Media Marketing. We use the data in June 2016, when the referendum for the UK's withdrawal from the EU (Brexit) was held on the 23rd of the month. The result of the Brexit referendum was announced during the trading hours of the TSE on that day. That news made the Japanese Stock Market plunge significantly. The Brexit referendum is arguably one of the biggest financial events in recent years. We can thus analyze the effect of the Brexit referendum on the volatility of the Japanese stock market. Another reason for this choice is that Japan has no holiday in June, so all weekdays are trading days. There are five weeks in June 2016. Since the first week of June 2016 includes 30 and 31 May and the last week includes 1 July, we also include them in the sample period.

The morning session of TSE starts at 9:00 a.m. and ends at 11:30 a.m. while the afternoon session of TSE starts at 12:30 a.m. and ends at 3:00 p.m., so both sessions last for 150 min. We treat the morning session and the afternoon session as if they are separated trading hours, and normalize the time stamps so that they take values within $[0, 1]$. As a result, $t = 0$ corresponds to 9:00 a.m. for the morning session, while it corresponds to 12:30 a.m. for the afternoon session. In the same manner, $t = 1$ corresponds to 11:30 a.m. for the morning session, while it corresponds to 3:00 p.m. for the afternoon session. In this empirical study, we estimate the Bernstein polynomial of the intraday seasonality in each session by allowing $\beta$ in (2) to differ from session to session.

We pick prices at every 1 min and compute 1 min log difference of prices as 1 min stock returns. Thus, the number of observations per session is 150. Furthermore, we put together all series of 1 min returns in each week. As a result, the total number of observations per week is $150 \times 2 \times 5 = 1500$. In addition, to simplify the interpretation of the estimation results, we standardize each week-long series of 1 min returns so that the sample mean is 0 and the sample variance is 1. Table 1 shows the descriptive statistics of the standard 1 min returns of TOPIX in each week, while Figures 1–5 show the time series plots of the standardized 1 min returns for each week.

**Table 1.** Descriptive statistics of standardized TOPIX 1 min returns in June 2016.

| Date | Skewness | Kurtosis | Min. | Max |
|---|---|---|---|---|
| Week 1 | −0.1081 | 7.2296 | −7.2569 | 5.4520 |
| Week 2 | 0.2494 | 7.7468 | −5.9886 | 5.7911 |
| Week 3 | 0.3534 | 7.4500 | −6.4415 | 5.8413 |
| Week 4 | −0.0125 | 7.1031 | −6.4212 | 5.6074 |
| Week 5 | 0.0346 | 4.9146 | −4.6433 | 5.4437 |

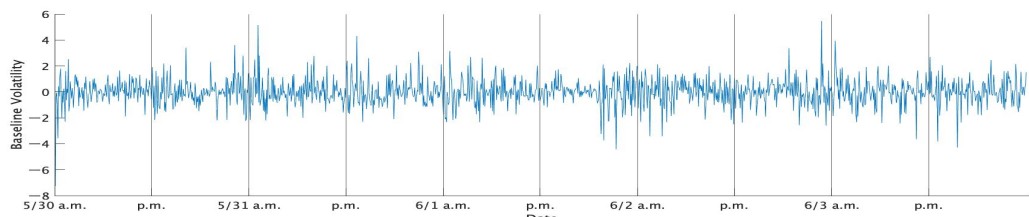

**Figure 1.** Standardized returns of the TOPIX data in the first week of June 2016.

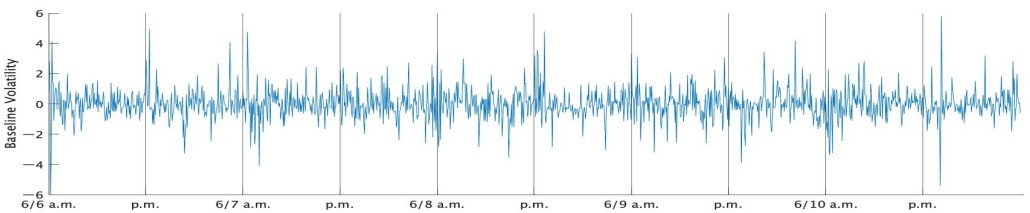

**Figure 2.** Standardized returns of the TOPIX data in the second week of June 2016.

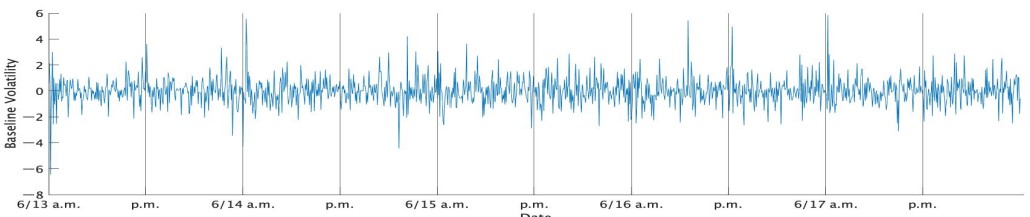

**Figure 3.** Standardized returns of the TOPIX data in the third week of June 2016.

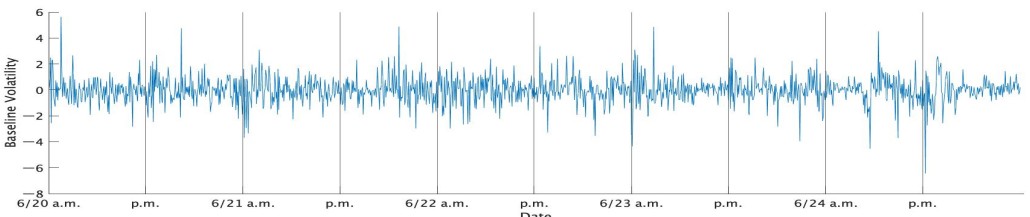

**Figure 4.** Standardized returns of the TOPIX data in the fourth week of June 2016.

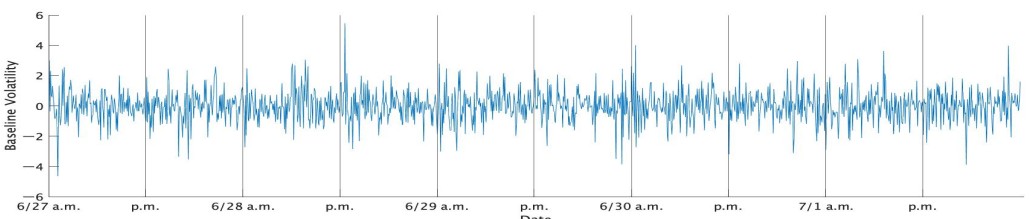

**Figure 5.** Standardized returns of the TOPIX data in the fifth week of June 2016.

We consider five candidates (SV-N, SV-G, SV-SG, SV-T, SV-ST) in the SV model (28) and set the prior distributions as follows:

$$\beta \sim \text{Normal}(0, 100I), \quad \gamma \sim \text{Normal}(0, 100), \quad \tau^2 \sim \text{Inverse Gamma}(1, 0.04),$$

$$\frac{\phi + 1}{2} \sim \text{Beta}(1, 1), \quad \alpha \sim \text{Normal}(0, 100), \quad \nu \sim \text{Gamma}(0, 0.1).$$

We vary the order of the Bernstein polynomial from 5 to 10. In sum, we try 30 different model specifications for the SV model (28). In the MCMC implementation, we generate 10,000 draws after the first 5000 draws are discarded as the burn-in periods. To select the best model among the candidates, we employ the widely applicable information criterion (WAIC, Watanabe 2010). We compute the WAIC of each model specification with the formula by Gelman et al. (2014). The results are reported in Tables 2–6. According to these tables, SV-G or SV-SG is the best model in all months. It may be a notable finding since the SV model with the variance-gamma error has hardly been applied in the previous studies.

**Table 2.** Widely applicable information criterion (WAIC) values of TOPIX returns (week 1).

|         | Order 5 | Order 6 | Order 7 | Order 8 | Order 9 | Order 10 |
|---------|---------|---------|---------|---------|---------|----------|
| SV-N    | 3965.1  | 3966.0  | 3962.0  | 3975.1  | 3969.6  | 3971.3   |
| SV-G    | 3701.0  | 3698.5  | 3702.2  | **3697.4** | 3700.0 | 3701.1  |
| SV-SG   | 3701.9  | 3701.0  | 3698.9  | 3699.0  | 3702.8  | 3700.8   |
| SV-T    | 3813.2  | 3813.3  | 3813.8  | 3816.6  | 3813.9  | 3813.0   |
| SV-ST   | 3819.3  | 3813.8  | 3816.5  | 3816.2  | 3815.7  | 3817.5   |

Note: Bold highlight means the best model according to WAIC.

**Table 3.** WAIC values of TOPIX returns (week 2).

|         | Order 5 | Order 6 | Order 7 | Order 8 | Order 9 | Order 10 |
|---------|---------|---------|---------|---------|---------|----------|
| SV-N    | 3811.0  | 3813.7  | 3814.5  | 3815.7  | 3818.0  | 3819.6   |
| SV-G    | 3621.0  | 3622.7  | 3621.0  | 3619.7  | 3621.2  | 3621.8   |
| SV-SG   | **3618.6** | 3621.3 | 3622.0 | 3623.7 | 3623.4 | 3619.7  |
| SV-T    | 3685.4  | 3684.1  | 3681.5  | 3859.0  | 3860.4  | 3684.6   |
| SV-ST   | 3686.9  | 3684.9  | 3683.3  | 3684.1  | 3684.3  | 3686.4   |

Note: Bold highlight means the best model according to WAIC.

**Table 4.** WAIC values of TOPIX returns (week 3).

|         | Order 5 | Order 6 | Order 7 | Order 8 | Order 9 | Order 10 |
|---------|---------|---------|---------|---------|---------|----------|
| SV-N    | 3976.9  | 3991.1  | 3996.8  | 3975.1  | 3969.6  | 3971.3   |
| SV-G    | **3750.2** | 3752.4 | 3752.0 | 3755.2 | 3752.9 | 3754.3  |
| SV-SG   | 3753.6  | 3753.3  | 3755.8  | 3754.1  | 3759.9  | 3752.8   |
| SV-T    | 3862.0  | 3859.4  | 3861.6  | 3859.0  | 3860.4  | 3861.6   |
| SV-ST   | 3862.1  | 3861.2  | 3861.3  | 3860.6  | 3861.5  | 3862.1   |

Note: Bold highlight means the best model according to WAIC.

**Table 5.** WAIC values of TOPIX returns (week 4).

|         | Order 5 | Order 6 | Order 7 | Order 8 | Order 9 | Order 10 |
|---------|---------|---------|---------|---------|---------|----------|
| SV-N    | 3927.5  | 3924.5  | 3924.8  | 3925.8  | 3924.7  | 3927.1   |
| SV-G    | 3670.3  | 3680.1  | 3679.0  | **3662.8** | 3675.7 | 3670.4  |
| SV-SG   | 3663.5  | 3668.2  | 3670.3  | 3664.6  | 3672.6  | 3669.1   |
| SV-T    | 3750.9  | 3751.9  | 3752.3  | 3750.8  | 3753.1  | 3753.9   |
| SV-ST   | 3753.5  | 3754.3  | 3753.4  | 3755.3  | 3752.8  | 3751.7   |

Note: Bold highlight means the best model according to WAIC.

**Table 6.** WAIC values of TOPIX returns (week 5).

|  | Order 5 | Order 6 | Order 7 | Order 8 | Order 9 | Order 10 |
|---|---|---|---|---|---|---|
| SV-N | 4114.6 | 4114.3 | 4111.3 | 4115.2 | 4113.7 | 4114.5 |
| SV-G | 3961.2 | 3959.6 | 3958.3 | 3960.0 | 3961.2 | 3960.0 |
| SV-SG | 3953.7 | 3955.6 | 3960.0 | 3963.3 | 3957.7 | **3953.1** |
| SV-T | 4033.7 | 4033.8 | 4032.7 | 4031.2 | 4032.8 | 4033.1 |
| SV-ST | 4032.5 | 4033.0 | 4033.4 | 4030.4 | 4032.0 | 4031.5 |

Note: Bold highlight means the best model according to WAIC.

For the selected models, we compute the posterior statistics (posterior means, standard deviations, 95% credible intervals and inefficiency factors) of the parameters and report them in Tables 7–11. The inefficiency factor measures how inefficient the MCMC sampling algorithm is (see e.g., Chib 2001). In these tables, the 95% credible intervals of the leverage parameter $\gamma$ and the asymmetric parameter $\alpha$ contain 0 for all specifications. Thus, we may conclude that the error distribution of 1 min returns of TOPIX is not asymmetric. In addition, most of the marginal posterior distributions of $\phi$ are concentrated near 1, even though the uniform prior is assumed for $\phi$. This suggests that the latent log volatility is strongly persistent, which is consistent with findings by previous studies on the stock markets. Regarding the tail parameter $\nu$, its marginal posterior distribution is centered around 2–6 in most models, which indicates that the excess kurtosis of the error distribution is high.

As for the intraday seasonality, the estimates of $\beta$ themselves are not of our interest. Instead we show the posterior mean and the 95% credible interval of the Bernstein polynomial $x_t'\beta$ in Figure 6. These figures show that some of the trading days exhibit the well-known U-shaped curve of intraday volatility, but others slant upward or downward. At the beginning on the day of Brexit (23 June), the market began with a highly volatile situation, but the volatility gradually became lower. During the afternoon session, the volatility was kept in a stable condition.

**Table 7.** Estimation results for TOPIX returns (week 1).

|  | $\gamma$ | $\phi$ | $\tau$ | $\alpha$ | $\nu$ |
|---|---|---|---|---|---|
| SV-N (7) [a] | -0.5742 [b] | 0.8909 | 0.1903 |  |  |
|  | $[-1.4270, -0.1241]$ [c] | $[0.8214, 0.9431]$ | $[0.1306, 0.2494]$ |  |  |
|  | 3.42 [d] | 4.01 | 4.54 |  |  |
| SV-G (8) | $-1.3565$ | 0.9608 | 0.0798 |  | 2.5248 |
|  | $[-3.1743, 0.8347]$ | $[0.9320, 0.9815]$ | $[0.0604, 0.1034]$ |  | $[2.0444, 3.2647]$ |
|  | 4.13 | 3.17 | 4.45 |  | 3.40 |
| SV-SG (7) | $-1.4520$ | 0.9598 | 0.0812 | 0.0014 | 2.5489 |
|  | $[-3.2233, 0.4995]$ | $[0.9316, 0.9802]$ | $[0.0630, 0.1077]$ | $[-0.0504, 0.0544]$ | $[2.0425, 3.3462]$ |
|  | 3.99 | 3.01 | 4.45 | 1.23 | 3.48 |
| SV-T (10) | $-1.4209$ | 0.9864 | 0.0766 |  | 4.2950 |
|  | $[-4.3298, 1.5726]$ | $[0.9745, 0.9955]$ | $[0.0594, 0.1010]$ |  | $[3.3294, 5.5097]$ |
|  | 3.85 | 2.90 | 4.42 |  | 3.44 |
| SV-ST (6) | $-1.6137$ | 0.9870 | 0.0753 | 0.0006 | 4.2338 |
|  | $[-5.0449, 1.6958]$ | $[0.9751, 0.9957]$ | $[0.0586, 0.0962]$ | $[-0.0400, 0.0407]$ | $[3.3247, 5.5254]$ |
|  | 3.94 | 2.78 | 4.38 | 1.62 | 3.54 |

a: the selected Bernstein polynomial order. b: posterior mean. c: 95% credible interval. d: inefficiency factor.

**Table 8.** Estimation results for TOPIX returns (week 2).

| | $\gamma$ | $\phi$ | $\tau$ | $\alpha$ | $\nu$ |
|---|---|---|---|---|---|
| SV-N (5) [a] | −0.2127 [b] [−0.5550, 0.0923] [c] 3.04 [d] | 0.7149 [0.5967, 0.8106] 3.87 | 0.3435 [0.2750, 0.4187] 4.39 | | |
| SV-G (8) | 0.0397 [−1.7042, 2.0695] 4.27 | 0.9191 [0.8122, 0.9694] 4.05 | 0.0917 [0.0604, 0.1442] 4.58 | | 2.2475 [2.0089, 2.7055] 2.6736 |
| SV-SG (5) | 0.0380 [−1.5878, 1.9299] 4.24 | 0.8965 [0.6761, 0.9664] 4.46 | 0.0991 [0.0647, 0.1610] 4.61 | −0.0001 [−0.0530, 0.0531] 1.24 | 2.2622 [2.0102, 2.7308] 2.85 |
| SV-T (7) | −0.7862 [−4.7784, 2.8361] 3.91 | 0.914 [0.9825, 0.9979] 2.83 | 0.0673 [0.0511, 0.0886] 4.43 | | 3.30 [2.7141, 4.0299] 3.17 |
| SV-ST (7) | −0.6862 [−4.4268, 2.9433] 3.93 | 0.9909 [0.9816, 0.9976] 2.77 | 0.0690 [0.0535, 0.0902] 4.42 | −0.0027 [−0.0376, 0.0323] 1.50 | 3.3647 [2.7611, 4.0958] 3.14 |

a: the selected Bernstein polynomial order. b: posterior mean. c: 95% credible interval. d: inefficiency factor.

**Table 9.** Estimation results for TOPIX returns (week 3).

| | $\gamma$ | $\phi$ | $\tau$ | $\alpha$ | $\nu$ |
|---|---|---|---|---|---|
| SV-N (9) [a] | 0.0514 [b] [−0.3345, 0.4436] [c] 2.71 [d] | 0.6249 [0.3506, 0.8190] 4.39 | 0.2950 [0.2042, 0.3872] 4.54 | | |
| SV-G (5) | 0.0639 [−1.8134, 1.9573] 4.20 | 0.4533 [0.1344, 0.7393] 4.33 | 0.0919 [0.0632, 0.1413] 4.57 | | 2.2888 [2.0155, 2.7501] 2.65 |
| SV-SG (10) | −0.2023 [−2.1985, 1.7331] 4.20 | 0.7992 [0.2317, 0.9511] 4.59 | 0.0851 [0.0596, 0.1250] 4.55 | −0.0031 [−0.0546, 0.0485] 1.17 | 2.3419 [2.0232, 2.9008] 2.99 |
| SV-T (8) | −0.2369 [−3.6943, 3.4453] 3.86 | 0.9871 [0.9755, 0.9960] 2.83 | 0.0661 [0.0514, 0.0837] 4.39 | | 4.0539 [3.2156, 5.1327] 3.39 |
| SV-ST (8) | −0.3313 [−4.2240, 3.6835] 3.98 | 0.9866 [0.9730, 0.9957] 3.07 | 0.0670 [0.0521, 0.0900] 4.42 | −0.0034 [−0.0426, 0.0361] 1.59 | 4.1237 [3.2114, 5.2538] 3.44 |

a: the selected Bernstein polynomial order. b: posterior mean. c: 95% credible interval. d: inefficiency factor.

**Table 10.** Estimation results for TOPIX returns (week 4).

| | $\gamma$ | $\phi$ | $\tau$ | $\alpha$ | $\nu$ |
|---|---|---|---|---|---|
| SV-N (6) [a] | −0.6502 [b] | 0.9336 | 0.1526 | | |
| | [−1.8098, 0.2658] [c] | [0.8831, 0.9704] | [0.1038, 0.2117] | | |
| | 3.53 [d] | 4.01 | 4.57 | | |
| SV-G (8) | −1.4435 | 0.9689 | 0.0853 | | 2.9487 |
| | [−3.6134, 0.6862] | [0.9454, 0.9859] | [0.0657, 0.1098] | | [2.1879, 3.9161] |
| | 4.11 | 3.10 | 4.43 | | 3.63 |
| SV-SG (5) | −1.7146 | 0.9694 | 0.0846 | −0.0068 | 2.93 |
| | [−4.0432, −0.4361] | [0.9458, 0.9868] | [0.0650, 0.1116] | [−0.0599, 0.0471] | [2.14, 3.94] |
| | 4.18 | 3.11 | 4.46 | 1.54 | 3.64 |
| SV-T (8) | −1.4043 | 0.9869 | 0.0824 | | 4.5805 |
| | [−4.4147, −1.6809] | [0.9747, 0.9960] | [0.0634, 0.1060] | | [3.5659, 5.8472] |
| | 3.96 | 2.89 | 4.41 | | 3.43 |
| SV-ST (10) | −1.6482 | 0.9882 | 0.0788 | −0.0045 | 4.4738 |
| | [−5.1451, 1.6390] | [0.9777, 0.9964] | [0.0623, 0.0982] | [−0.0460, 0.0362] | [3.4944, 5.7573] |
| | 4.05 | 2.64 | 4.36 | 1.70 | 3.39 |

a: the selected Bernstein polynomial order. b: posterior mean. c: 95% credible interval. d: inefficiency factor.

**Table 11.** Estimation results for TOPIX returns (week 5).

| | $\gamma$ | $\phi$ | $\tau$ | $\alpha$ | $\nu$ |
|---|---|---|---|---|---|
| SV-N (7) [a] | −0.2602 [b] | 0.8529 | 0.1428 | | |
| | [−1.5977, 0.8891] [c] | [0.6617, 0.9395] | [0.0858, 0.2348] | | |
| | 3.47 [d] | 4.38 | 4.62 | | |
| SV-G (7) | −1.1175 | 0.8638 | 0.0873 | | 3.6578 |
| | [−3.7543, 1.4340] | [0.6609, 0.9457] | [0.0642, 0.1144] | | [2.4812, 5.1414] |
| | 4.21 | 4.24 | 4.45 | | 3.86 |
| SV-SG (10) | −1.0562 | 0.8226 | 0.0828 | −0.0042 | 3.5728 |
| | [−4.0335, 2.0096] | [0.1974, 0.9485] | [0.0587, 0.1170] | [−0.0565, 0.0490] | [2.4627, 4.8738] |
| | 4.27 | 4.60 | 4.52 | 1.19 | 3.73 |
| SV-T (8) | −1.2853 | 0.9727 | 0.0730 | | 6.2435 |
| | [−4.7632, 1.8139] | [0.9448, 0.9898] | [0.0547, 0.0998] | | [4.5496, 8.6440] |
| | 3.92 | 3.59 | 4.50 | | 3.71 |
| SV-ST (8) | −1.5744 | 0.9726 | 0.0735 | −0.0084 | 6.2388 |
| | [−5.5186, 1.9908] | [0.9459, 0.9898] | [0.0566, 0.0992] | [−0.0536, 0.0378] | [4.6016, 8.6696] |
| | 4.04 | 3.56 | 4.46 | 1.84 | 3.74 |

a: the selected Bernstein polynomial order. b: posterior mean. c: 95% credible interval. d: inefficiency factor.

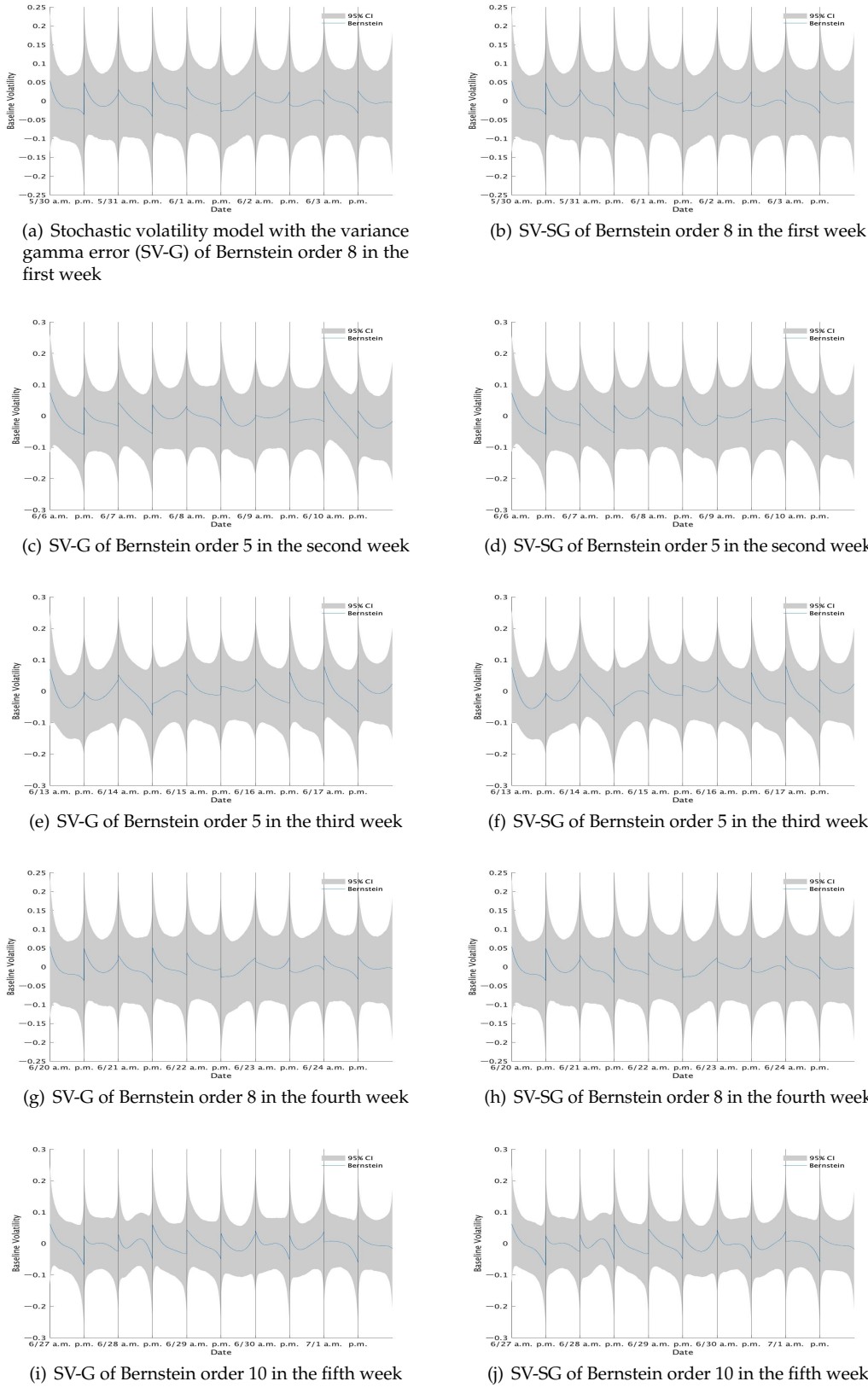

(a) Stochastic volatility model with the variance gamma error (SV-G) of Bernstein order 8 in the first week

(b) SV-SG of Bernstein order 8 in the first week

(c) SV-G of Bernstein order 5 in the second week

(d) SV-SG of Bernstein order 5 in the second week

(e) SV-G of Bernstein order 5 in the third week

(f) SV-SG of Bernstein order 5 in the third week

(g) SV-G of Bernstein order 8 in the fourth week

(h) SV-SG of Bernstein order 8 in the fourth week

(i) SV-G of Bernstein order 10 in the fifth week

(j) SV-SG of Bernstein order 10 in the fifth week

**Figure 6.** Intraday seasonality with Bernstein polynomial approximation.

## 5. Conclusions

In this paper, we extended the standard SV model into a more general formulation so that it could capture key characteristics of intraday high-frequency stock returns such as intraday seasonality, asymmetry and excess kurtosis. Our proposed model uses a Bernstein polynomial of time stamps as the intraday seasonal component of the stock volatility, and the coefficients in the Bernstein polynomial are simultaneously estimated along with the rest of the parameters in the model. To incorporate asymmetry and excess kurtosis into the standard SV model, we assume that the error distribution of stock returns in the SV model belongs to a family of generalized hyperbolic distributions. In particular, we focus on two sub-classes of this family: skew Student's *t* distribution and skew variance-gamma distribution. Furthermore we developed an efficient MCMC sampling algorithm for Bayesian inference on the proposed model by utilizing all without a loop (AWOL), ASIS and the generalized Gibbs sampler.

As an application, we estimated the proposed SV models with 1 min return data of TOPIX in various specifications and conducted model selection with WAIC. The model selection procedure chose the SV model with the variance-gamma-type error as the most suitable one. The estimated parameters indicated strong excess kurtosis in the error distribution of 1 min returns, though the asymmetry was not supported since both leverage parameter $\gamma$ and asymmetry parameter $\alpha$ were not significantly different from zero. Furthermore our proposed model successfully extracted intraday seasonal patterns in the stock volatility with Bernstein polynomial approximation, though the shape of the intraday seasonal component was not necessarily U-shaped.

**Author Contributions:** M.N. had full access to the data in the study and takes responsibility for the accuracy and integrity of the data and its analyses. Study concept and design: All authors. Acquisition, analysis, or interpretation of data: All authors. Drafting of the manuscript: M.N. Critical revision of the manuscript for important intellectual content: T.N. Statistical analysis: All authors. Administrative, technical or material support: All authors. Study supervision: T.N. All authors have read and agreed to the published version of the manuscript

**Funding:** This study was funded by the Japan Society for the Promotion of Science (JSPS) KAKENHI Grant Number 19K01592.

**Institutional Review Board Statement:** Not applicable.

**Informed Consent Statement:** Not applicable.

**Data Availability Statement:** Not applicable.

**Conflicts of Interest:** The authors declare no conflict of interest.

## Appendix A. Conditional Posterior Distributions

In Appendix, we derive the conditional posterior distribution of the latent log volatility and that of each parameter in the SV model for both NCP and CP.

*Appendix A.1. NCP Form*

Appendix A.1.1. Latent Log Volatility $h_{1:T+1}$

The conditional posterior density of the latent log volatility $h_{1:T+1}$ is

$$p(h_{1:T+1}|\theta, y_{1:T}) \propto \prod_{t=1}^{T} p(y_t|h_t, h_{t+1}, \theta) \cdot p(h_{1:T+1}|\theta). \tag{A1}$$

We apply the Metropolis-Hastings (MH) algorithm to generate $h_{1:T+1}$ from (A1). To derive a suitable proposal distribution for the MH algorithm, we first consider consider the second-

order Taylor approximation of $\ell(h_{1:T+1}) = \log p(y_{1:T}|h_{1:T+1}, \theta)$ in the neighborhood of $h_{1:T+1}^*$:

$$
\ell(h_{1:T+1}) \approx \ell(h_{1:T+1}^*) + g(h_{1:T+1}^*)'(h_{1:T+1} - h_{1:T+1}^*)
$$
$$
- \frac{1}{2}(h_{1:T+1} - h_{1:T+1}^*)'Q(h_{1:T+1}^*)(h_{1:T+1} - h_{1:T+1}^*), \tag{A2}
$$

where $g(h_{1:T+1})$ is the gradient vector of $\ell(h_{1:T+1})$:

$$
g(h_{1:T+1}) = \begin{bmatrix} g_1(h_{1:T+1}) \\ \vdots \\ g_t(h_{1:T+1}) \\ \vdots \\ g_T(h_{1:T+1}) \\ g_{T+1}(h_{1:T+1}) \end{bmatrix} = \begin{bmatrix} \nabla_1 \log p(y_{1:T}|h_{1:T+1}, \theta) \\ \vdots \\ \nabla_t \log p(y_{1:T}|h_{1:T+1}, \theta) \\ \vdots \\ \nabla_T \log p(y_{1:T}|h_{1:T+1}, \theta) \\ \nabla_{T+1} \log p(y_{1:T}|h_{1:T+1}, \theta) \end{bmatrix},
$$

and $Q(h_{1:T+1})$ is the Hessian matrix of $\log p(y_{1:T}|h_{1:T+1}, \theta)$ times $-1$:

$$
Q(h_{1:T+1}) = \begin{bmatrix} q_{11}(h_{1:T+1}) & q_{12}(h_{1:T+1}) & & \cdots & 0 \\ q_{21}(h_{1:T+1}) & q_{22}(h_{1:T+1}) & q_{23}(h_{1:T+1}) & \cdots & 0 \\ \vdots & \ddots & \ddots & \ddots & \vdots \\ 0 & \cdots & q_{T,T-1}(h_{1:T+1}) & q_{T,T}(h_{1:T+1}) & q_{T,T+1}(h_{1:T+1}) \\ 0 & \cdots & & q_{T+1,T}(h_{1:T+1}) & q_{T+1,T+1}(h_{1:T+1}) \end{bmatrix}.
$$

which is a $(T+1) \times (T+1)$ band matrix.

Let us derive the explicit form of each element in $g(h_{1:T+1})$ and $Q(h_{1:T+1})$. By defining

$$
\epsilon_t = y_t \exp(-x_t'\beta - h_t), \quad \eta_t = h_{t+1} - \phi h_t, \tag{A3}
$$

the log density of $y_t$ (7) is rewritten as

$$
\log p(y_t|h_t, h_{t+1}, \theta) = -x_t'\beta - h_t - \frac{1}{2}(\epsilon_t - \gamma\eta_t)^2 + \text{constant.}
$$

Note that

$$
\nabla_t \epsilon_t = -\epsilon_t, \quad \nabla_t \eta_t = -\phi, \quad \nabla_t \eta_{t-1} = 1.
$$

where $\nabla_t = \frac{\partial}{\partial h_t}$. Each element of $g(h_{1:T+1})$ is derived as

$$
\begin{aligned}
g_t(h_{1:T+1}) &= \nabla_t \log p(y_{1:T}|h_{1:T+1}, \theta) = \nabla_t \log p(y_t|h_t, h_{t+1}, \theta) + \nabla_t \log p(y_{t-1}|h_{t-1}, h_t, \theta) \\
&= -1 - (\epsilon_t - \gamma\eta_t)(-\epsilon_t - \gamma(-\phi)) - (\epsilon_{t-1} - \gamma\eta_{t-1})(-\gamma) \\
&= -1 + (\epsilon_t - \gamma\eta_t)(\epsilon_t - \gamma\phi) + \gamma(\epsilon_{t-1} - \gamma\eta_{t-1}),
\end{aligned}
$$

for $t = 2, \ldots, T$,

$$
\begin{aligned}
g_1(h_{1:T+1}) &= \nabla_1 \log p(y_{1:T}|h_{1:T+1}, \theta) = \nabla_1 \log p(y_1|h_1, h_2, \theta) \\
&= -1 - (\epsilon_1 - \gamma\eta_1)(-\epsilon_1 - \gamma(-\phi)) \\
&= -1 + (\epsilon_1 - \gamma\eta_1)(\epsilon_1 - \gamma\phi),
\end{aligned}
$$

for $t = 1$, and

$$
\begin{aligned}
g_{T+1}(h_{1:T+1}) &= \nabla_{T+1} \log p(y_{1:T}|h_{1:T+1}, \theta) = \nabla_{T+1} \log p(y_T|h_T, h_{T+1}, \theta) \\
&= -(\epsilon_T - \gamma\eta_T)(-\gamma) = \gamma(\epsilon_T - \gamma\eta_T),
\end{aligned}
$$

for $t = T + 1$. The diagonal element in $Q(h_{1:T+1})$ is given as

$$
\begin{aligned}
q_{t,t}(h_{1:T+1}) &= (-1) \times \nabla_t^2 \log p(y_{1:T}|h_{1:T+1}, \theta) \\
&= -(-\epsilon_t - \gamma(-\phi))(\epsilon_t - \gamma\phi) - (\epsilon_t - \gamma\eta_t)(-\epsilon_t) - \gamma(-\gamma) \\
&= (\epsilon_t - \gamma\phi)^2 + \epsilon_t(\epsilon_t - \gamma\eta_t) + \gamma^2,
\end{aligned}
$$

for $t = 2, \ldots, T$,

$$
\begin{aligned}
q_{11}(h_{1:T+1}) &= (-1) \times \nabla_1^2 \log p(y_{1:T}|h_{1:T+1}, \theta) \\
&= -(-\epsilon_1 - \gamma(-\phi))(\epsilon_1 - \gamma\phi) - (\epsilon_1 - \gamma\eta_1)(-\epsilon_1) \\
&= (\epsilon_1 - \gamma\phi)^2 + \epsilon_1(\epsilon_1 - \gamma\eta_1),
\end{aligned}
$$

for $t = 1$, and

$$
\begin{aligned}
q_{T+1,T+1}(h_{1:T+1}) &= (-1) \times \nabla_{T+1}^2 \log p(y_{1:T}|h_{1:T+1}, \theta) \\
&= -\gamma(-\gamma) = \gamma^2,
\end{aligned}
$$

for $t = T + 1$. Furhtermore the first off-diagonal element of $Q(h_{1:T+1})$ is derived as

$$
\begin{aligned}
q_{t,t+1}(h_{1:T+1}) &= (-1) \times \nabla_{t,t+1} \log p(y_{1:T}|h_{1:T+1}, \theta) \\
&= -(-\gamma)(\epsilon_t - \gamma\phi) \\
&= \gamma(\epsilon_t - \gamma\phi),
\end{aligned}
$$

for $t = 1, \ldots, T$. In summary,

$$g_t(h_{1:T+1}) = \{-1 + (\epsilon_t - \gamma\eta_t)(\epsilon_t - \gamma\phi)\}\mathbf{1}(t \leqq T) + \gamma(\epsilon_{t-1} - \gamma\eta_{t-1})\mathbf{1}(t \geqq 2), \quad \text{(A4)}$$
$$q_t(h_{1:T+1}) = \{(\epsilon_t - \gamma\phi)^2 + \epsilon_t(\epsilon_t - \gamma\eta_t)\}\mathbf{1}(t \leqq T) + \gamma^2\mathbf{1}(t \geqq 2), \quad \text{(A5)}$$
$$q_{t,t+1}(h_{1:T+1}) = \gamma(\epsilon_t - \gamma\phi). \quad \text{(A6)}$$

Since the log prior density of $h_{1:T+1}$ is

$$\bar{p}(h_{1:T+1}) = -\frac{T+1}{2}\log(2\pi\tau^2) + \frac{1}{2}\log|V| - \frac{1}{2\tau^2}h'_{1:T+1}Vh_{1:T+1}, \quad \text{(A7)}$$

the conditional posterior density of $h_{1:T+1}$ (A1) can be approximated by

$$
\begin{aligned}
&p(h_{1:T+1}|\theta, y_{1:T}) \\
&= C\exp[\ell(h_{1:T+1}) + \bar{p}(h_{1:T+1})] \\
&\approx C\exp\Big[\ell(h^*_{1:T+1}) + g(h^*_{1:T+1})'(h_{1:T+1} - h^*_{1:T+1}) \\
&\qquad\quad - \frac{1}{2}(h_{1:T+1} - h^*_{1:T+1})'Q(h^*_{1:T+1})(h_{1:T+1} - h^*_{1:T+1}) + \bar{p}(h_{1:T+1})\Big] \\
&= C\exp\Big[\ell(h^*_{1:T+1}) - \frac{T+1}{2}\log(2\pi\tau^2) + \frac{1}{2}\log|V| + f(h_{1:T+1})\Big], \quad \text{(A8)}
\end{aligned}
$$

where $C$ is the normalizing constant of the conditional posterior density and

$$
\begin{aligned}
f(h_{1:T+1}) &= g(h^*_{1:T+1})'(h_{1:T+1} - h^*_{1:T+1}) \\
&\quad - \frac{1}{2}(h_{1:T+1} - h^*_{1:T+1})'Q(h^*_{1:T+1})(h_{1:T+1} - h^*_{1:T+1}) \\
&\quad - \frac{1}{2\tau^2}h'_{1:T+1}Vh_{1:T+1}, \quad \text{(A9)}
\end{aligned}
$$

By completing the square in (A9), we have

$$f(h_{1:T+1}) = -\frac{1}{2}\big(h_{1:T+1} - \mu_h(h^*_{1:T+1})\big)'\Sigma_h(h^*_{1:T+1})^{-1}\big(h_{1:T+1} - \mu_h(h^*_{1:T+1})\big) \tag{A10}$$
$$+ \text{constant},$$

where

$$\Sigma_h(h^*_{1:T+1}) = \left(Q(h^*_{1:T+1}) + \frac{1}{\tau^2}V\right)^{-1},$$
$$\mu_h(h^*_{1:T+1}) = \Sigma_h(h^*_{1:T+1})\big(g(h^*_{1:T+1}) + Q(h^*_{1:T+1})h^*_{1:T+1}\big).$$

Therefore the right-hand side of (A8) is approximately proportional to the pdf of the following normal distribution:

$$h_{1:T+1} \sim \text{Normal}\big(\mu_h(h^*_{1:T+1}), \Sigma_h(h^*_{1:T+1})\big). \tag{A11}$$

Recall that both $Q(h^*_{1:T+1})$ and $V$ are tridiagonal matrices. Thus, $\Sigma_h(h^*_{1:T+1})^{-1} = Q(h^*_{1:T+1}) + \frac{1}{\tau^2}V$ is also tridiagonal. Since the Cholesky decomposition of a tridiagonal matrix and the inverse of a triangular matrix can be efficiently computed if they exist, $h_{1:T+1}$ is readily generated from (A11) with

$$h_{1:T+1} = \big(L'\big)^{-1}\Big(L^{-1}\big(g(h^*_{1:T+1}) + Q(h^*_{1:T+1})h^*_{1:T+1}\big) + \tilde{z}\Big), \quad \tilde{z} \sim \text{Normal}(0, I),$$

where $L$ is a lower triangular matrix obtained by the Cholesky decomposition as

$$L'L = Q(h^*_{1:T+1}) + \frac{1}{\tau^2}V.$$

The above algorithm, which is called the *all without a loop* (AWOL) in Kastner and Frühwirth-Schnatter (2014), has been applied to Gaussian Markov random fields (e.g., Rue 2001) and state-space models (e.g., Chan and Jeliazkov 2009; McCausland et al. 2011).

Hoping that the approximation (A8) is sufficiently accurate, we use (A11) as the proposal distribution in the MH algorithm. In practice, however, we need to address two issues:

1. the choice of $h^*_{1:T+1}$ is crucial to make the approximation (A8) workable.
2. the acceptance rate of the MH algorithm tends to be too low when $h_{1:T+1}$ is a high-dimensional vector.

We address the former issue by using the mode of the conditional posterior density as $h^*_{1:T+1}$. The search of the mode is performed by the following recursion:

**Step 1:** Initialize $h^{*(0)}_{1:T+1}$ and set the counter $r = 1$.

**Step 2:** Update $h^{*(r)}_{1:T+1}$ by $h^{*(r)}_{1:T+1} = \mu_h(h^{*(r-1)}_{1:T+1})$.

**Step 3:** Let $r = r + 1$ and go to **Step 2** unless $\max_{t=1,\dots,T+1}|h^{*(r)}_t - h^{*(r-1)}_t|$ is less than the preset tolerance level.

In our experience, it mostly attains convergence in a few iterations.

We address the latter issue by applying a so-called block sampler. In the block sampler, we randomly partition $h_{1:T+1}$ into several sub-vectors (blocks), generate each block from its conditional distribution given the rest of the blocks and apply the MH algorithm to each generated block. Without loss of generality, suppose the proposal distribution (A11) is partitioned as

$$\underbrace{\begin{bmatrix} h_1 \\ h_2 \end{bmatrix}}_{h_{1:T}} \sim \text{Normal}\left(\underbrace{\begin{bmatrix} \mu_{h1} \\ \mu_{h2} \end{bmatrix}}_{\mu_h}, \underbrace{\begin{bmatrix} \Sigma_{h11} & \Sigma_{h12} \\ \Sigma_{h21} & \Sigma_{h11} \end{bmatrix}}_{\Sigma_h}\right), \tag{A12}$$

where $\mu_h = \mu_h(h^*_{1:T+1})$ and $\Sigma_h = \Sigma_h(h^*_{1:T+1})$ (we ignore the dependence on $h^*_{1:T+1}$ for brevity) and $h_1$ is the block to be updated in the current MH step, while $h_2$ contains either elements that were already updated in the previous MH steps or those to be updated in the following MH steps. It is well known that the conditional distribution of $h_1$ given $h_2$ is given by

$$h_1|h_2 \sim \text{Normal}\left(\mu_{h1} + \Sigma_{h11}\Sigma_{12}^{-1}(h_2 - \mu_{h2}), \Sigma_{h11} - \Sigma_{h12}\Sigma_{h22}^{-1}\Sigma_{h21}\right). \tag{A13}$$

Note that the inverse of the covariance matrix $\Sigma_h$ in (A12) is

$$\Sigma_h^{-1} = \begin{bmatrix} \Sigma_{h11} & \Sigma_{h12} \\ \Sigma_{h21} & \Sigma_{h22} \end{bmatrix}^{-1} = \begin{bmatrix} \Omega_{h11} & -\Omega_{h11}\Sigma_{h12}\Sigma_{h22}^{-1} \\ -\Sigma_{h22}^{-1}\Sigma_{h21}\Omega_{h11} & \Sigma_{h22}^{-1} + \Sigma_{h22}^{-1}\Sigma_{h21}\Omega_{h11}\Sigma_{h12}\Sigma_{h22}^{-1} \end{bmatrix},$$

$$\Omega_{h11} = \left(\Sigma_{h11} - \Sigma_{h12}\Sigma_{h22}^{-1}\Sigma_{h21}\right)^{-1}.$$

Furthermore, if we let $\Omega_{h12}$ denote the upper-right block of $\Sigma_h^{-1}$, we have

$$\Omega_{h12} = -\Omega_{h11}\Sigma_{h12}\Sigma_{h22}^{-1}.$$

Therefore the conditional distribution of $h_1$ given $h_2$ in (A13) is rearranged as

$$h_1|h_2 \sim \text{Normal}\left(\mu_{h1} - \Omega_{h11}^{-1}\Omega_{h12}(h_2 - \mu_{h2}), \Omega_{h11}^{-1}\right). \tag{A14}$$

Recall that $\Sigma_h^{-1}$ is tridiagonal and so is $\Omega_{h11}$ by construction. Thus, we can apply the AWOL algorithm:

$$h_1 = \mu_{h1} - \left(L_1'\right)^{-1}\left(L_1^{-1}\Omega_{h12}(h_2 - \mu_{h2}) - \tilde{z}_1\right), \quad \tilde{z}_1 \sim \text{Normal}(0, I), \quad L_1 L_1' = \Omega_{h11},$$

to generate $h_1$ from (A14). In essence, our approach is an AWOL variant of the block sampler proposed by Omori and Watanabe (2008).

Appendix A.1.2. Regression Coefficients $\beta$

The sampling scheme for the regression coefficients $\beta$ is almost identical to the one for the log volatility $h_{1:T+1}$. Let $\ell(\beta)$ denote $\log p(y_{1:T}|h_{1:T+1}, \theta)$ given $y_{1:T}$ and the parameters other than $\beta$. In the same manner as (A2), consider the second-order Taylor approximation of $\ell(\beta)$ in the neighborhood of $\beta^*$:

$$\ell(\beta) \approx \ell(\beta^*) + g(\beta^*)'(\beta - \beta^*) - \frac{1}{2}(\beta - \beta^*)'Q(\beta^*)(\beta - \beta^*), \tag{A15}$$

where $g(\beta)$ is the gradient vector of $\ell(\beta)$ and $Q(\beta)$ is the Hessian matrix of $\ell(\beta)$ times $-1$. Since $\nabla_\beta \epsilon_t = -\epsilon_t x_t$, we have

$$\nabla_\beta \log p(y_t|h_t, h_{t+1}, \theta) = -x_t + (\epsilon_t^2 - \gamma\eta_t\epsilon_t)x_t,$$
$$\nabla_\beta'\nabla_\beta \log p(y_t|h_t, h_{t+1}, \theta) = (-2\epsilon_t^2 + \gamma\eta_t\epsilon_t)x_t x_t'.$$

Therefore, $g(\beta)$ and $Q(\beta)$ are obtained as

$$g(\beta) = \sum_{t=1}^{T}(\epsilon_t(\epsilon_t - \gamma\eta_t) - 1)x_t,$$

$$Q(\beta) = \sum_{t=1}^{T}\epsilon_t(2\epsilon_t - \gamma\eta_t)x_t x_t'.$$

With the prior $\beta \sim \text{Normal}(\bar{\mu}_\beta, \bar{\Omega}_\beta^{-1})$, the conditional posterior density of $\beta$ can be approximated by

$$
\begin{aligned}
&p(\beta|h_{1:T+1}, \theta_{-\beta}, y_{1:T}) \\
&= C\exp[\ell(\beta) + \log p(\beta)] \\
&\approx C\exp\left[\ell(\beta^*) - \frac{1}{2}\log(2\pi) + \frac{1}{2}\log|\bar{\Omega}_\beta|\right] \\
&\quad \times \exp\left[g(\beta^*)'(\beta - \beta^*) - \frac{1}{2}(\beta - \beta^*)'Q(\beta^*)(\beta - \beta^*) - \frac{1}{2}(\beta - \bar{\mu}_\beta)'\bar{\Omega}_\beta(\beta - \bar{\mu}_\beta)\right].
\end{aligned} \quad (A16)
$$

By completing the square as in (A16), the proposal distribution for the MH algorithm is derived as

$$
\beta \sim \text{Normal}\left(\mu_\beta(\beta^*), \Sigma_\beta(\beta^*)\right), \quad (A17)
$$

where

$$
\Sigma_\beta(\beta^*) = \left(Q(\beta^*) + \bar{\Omega}_\beta\right)^{-1}, \quad \mu_\beta(\beta^*) = \Sigma_\beta(\beta^*)\left(g(\beta^*) + Q(\beta^*)\beta^* + \bar{\Omega}_\beta\bar{\mu}_\beta\right).
$$

The search algorithm for $\beta^*$ is the same as $h^*_{1:T+1}$.

Since the dimension of $\beta$ is considerably smaller than $h_{1:T+1}$, it is not necessary to apply the block sampler in our experience.

Appendix A.1.3. Leverage Parameter $\gamma$

Since we use the standard conditionally conjugate prior distributions for $\gamma$, the conditional posterior distribution is given by

$$
\gamma|h_{1:T+1}, \theta_{-\gamma}, y_{1:T} \sim \text{Normal}\left(\frac{\sum_{t=1}^{T}\eta_t\epsilon_t + \bar{\omega}_\gamma\bar{\mu}_\gamma}{\sum_{t=1}^{T}\eta_t^2 + \bar{\omega}_\gamma}, \frac{1}{\sum_{t=1}^{T}\eta_t^2 + \bar{\omega}_\gamma}\right). \quad (A18)
$$

Appendix A.1.4. Variance $\tau^2$

Since we use the standard conditionally conjugate prior distribution for $\tau^2$, the conditional posterior distribution is given by

$$
\tau^2|h_{1:T+1}, \theta_{-\tau^2}, y_{1:T} \sim \text{Inv. Gamma}\left(\frac{T+1}{2} + a_\tau, \frac{1}{2}h'_{1:T+1}Vh_{1:T+1} + b_\tau\right). \quad (A19)
$$

Appendix A.1.5. AR(1) Coefficient $\phi$

Once the state variables $h_{1:T+1}$ are generated, the conditional posterior density of $\phi$ is given by

$$
\begin{aligned}
p(\phi|h_{1:T+1}, \theta_{-\phi}, y_{1:T}) &\propto \sqrt{1 - \phi^2}\exp\left[-\frac{(1 - \phi^2)h_1^2 + \sum_{t=1}^{T}(h_{t+1} - \phi h_t)^2}{2\tau^2}\right] \\
&\quad \times (1 + \phi)^{a_\phi - 1}(1 - \phi)^{b_\phi - 1}\mathbf{1}_{(-1,1)}(\phi).
\end{aligned} \quad (A20)
$$

By completing the square, we have

$$(1 - \phi^2)h_1^2 + \sum_{t=1}^{T}(h_{t+1} - \phi h_t)^2$$

$$= (1 - \phi^2)h_1^2 + \sum_{t=1}^{T} h_{t+1}^2 - 2\phi \sum_{t=1}^{T} h_{t+1}h_t + \phi^2 \sum_{t=1}^{T} h_t^2$$

$$= \sum_{t=1}^{T+1} h_t^2 - 2\phi \sum_{t=1}^{T} h_{t+1}h_t + \phi^2 \sum_{t=2}^{T} h_t^2$$

$$= \sum_{t=2}^{T} h_t^2 \left(\phi - \frac{\sum_{t=1}^{T} h_{t+1}h_t}{\sum_{t=2}^{T} h_t^2}\right)^2 + \sum_{t=1}^{T+1} h_t^2 - \frac{\left(\sum_{t=1}^{T} h_{t+1}h_t\right)^2}{\sum_{t=2}^{T} h_t^2}.$$

With the above expression in mind, we use the following truncated normal distribution:

$$\phi \sim \text{Normal}\left(\frac{\sum_{t=1}^{T} h_{t+1}h_t}{\sum_{t=2}^{T} h_t^2}, \frac{\tau^2}{\sum_{t=2}^{T} h_t^2}\middle| -1 < \phi < 1\right), \tag{A21}$$

as the proposal distribution for $\phi$ in the MH algorithm.

*Appendix A.2. CP Form*

Appendix A.2.1. Latent Log Volatility $\tilde{h}_{1:T+1}$

The sampling scheme from the conditional posterior distribution of $\tilde{h}_{1:T+1}$:

$$p(\tilde{h}_{1:T+1}|\theta, y_{1:T}) \propto \prod_{t=1}^{T} p(y_t|\tilde{h}_t, \tilde{h}_{t+1}, \theta) \cdot p(\tilde{h}_{1:T+1}|\theta), \tag{A22}$$

is based on the MH algorithm, which is similar to the case of the NCP form. To construct the proposal distribution of $\tilde{h}_{1:T+1}$, we consider the second-order Taylor approximation of $\ell(\tilde{h}_{1:T+1}) = \log p(\tilde{h}_{1:T+1}|\theta, y_{1:T})$ in the neighborhood of $\tilde{h}_{1:T+1}^*$ as in (A2). We first derive the explicit form of each element in $g(\tilde{h}_{1:T+1})$ and $Q(\tilde{h}_{1:T+1})$. By defining

$$\tilde{\epsilon}_t = y_t \exp(-\tilde{h}_t), \quad \tilde{\eta}_t = \tilde{h}_{t+1} - \phi\tilde{h}_t - (x_{t+1} - \phi x_t)'\beta, \tag{A23}$$

the log density of $y_t$ in (15) is rewritten as

$$\log p(y_t|\tilde{h}_t, \tilde{h}_{t+1}, \theta) = -\tilde{h}_t - \frac{1}{2}(\tilde{\epsilon}_t - \gamma\tilde{\eta}_t)^2 + \text{constant}.$$

Since

$$\nabla_t \tilde{\epsilon}_t = -\tilde{\epsilon}_t, \quad \nabla_t \tilde{\eta}_t = -\phi, \quad \nabla_t \tilde{\eta}_{t-1} = 1,$$

$g_t(\tilde{h}_{1:T+1})$, $q_t(\tilde{h}_{1:T+1})$ and $q_{t,t+1}(\tilde{h}_{1:T+1})$ are identical to (A4)–(A6) except that $\epsilon_t$ and $\eta_t$ are replaced with $\tilde{\epsilon}_t$ and $\tilde{\eta}_t$, respectively.

Since the log prior density of $\tilde{h}_{1:T+1}$ is

$$\bar{p}(\tilde{h}_{1:T+1}) = -\frac{T+1}{2}\log(2\pi\tau^2) + \frac{1}{2}\log|V| - \frac{1}{2\tau^2}(\tilde{h}_{1:T+1} - X\beta)'V(\tilde{h}_{1:T+1} - X\beta), \tag{A24}$$

the conditional posterior density of $\tilde{h}_{1:T+1}$ (A22) can be approximated by

$$p(\tilde{h}_{1:T+1}|\theta, y_{1:T})$$
$$= C\exp[\ell(h_{1:T+1}) + \bar{p}(h_{1:T+1})]$$
$$\approx C\exp\left[\ell(\tilde{h}_{1:T+1}^*) - \frac{T+1}{2}\log(2\pi\tau^2) + \frac{1}{2}\log|V| + f(\tilde{h}_{1:T+1})\right], \tag{A25}$$

where $C$ is the normalizing constant of the conditional posterior density and

$$
\begin{aligned}
f(\tilde{h}_{1:T+1}) = {} & g(\tilde{h}^*_{1:T+1})'(\tilde{h}_{1:T+1} - \tilde{h}^*_{1:T+1}) \\
& - \frac{1}{2}(\tilde{h}_{1:T+1} - \tilde{h}^*_{1:T+1})'Q(\tilde{h}^*_{1:T+1})(\tilde{h}_{1:T+1} - \tilde{h}^*_{1:T+1}) \\
& - \frac{1}{2\tau^2}(\tilde{h}_{1:T+1} - X\beta)'V(\tilde{h}_{1:T+1} - X\beta),
\end{aligned}
\tag{A26}
$$

By completing the square in (A26), we have

$$
f(\tilde{h}_{1:T+1}) = -\frac{1}{2}(\tilde{h}_{1:T+1} - \mu_{\tilde{h}}(\tilde{h}^*_{1:T+1}))'\Sigma_{\tilde{h}}(\tilde{h}^*_{1:T+1})^{-1}(\tilde{h}_{1:T+1} - \mu_{\tilde{h}}(\tilde{h}^*_{1:T+1}))
\tag{A27}
$$
$$
+ \text{constant},
$$

where

$$
\Sigma_{\tilde{h}}(\tilde{h}^*_{1:T+1}) = \left(Q(\tilde{h}^*_{1:T+1}) + \frac{1}{\tau^2}V\right)^{-1},
$$

$$
\mu_{\tilde{h}}(\tilde{h}^*_{1:T+1}) = \Sigma_{\tilde{h}}(\tilde{h}^*_{1:T+1})\left(g(\tilde{h}^*_{1:T+1}) + Q(\tilde{h}^*_{1:T+1})\tilde{h}^*_{1:T+1} + \frac{1}{\tau^2}VX\beta\right).
$$

Therefore, the right-hand side of (A25) is approximately proportional to the pdf of the following normal distribution:

$$
\tilde{h}_{1:T+1} \sim \text{Normal}\left(\mu_{\tilde{h}}(\tilde{h}^*_{1:T+1}), \Sigma_{\tilde{h}}(\tilde{h}^*_{1:T+1})\right),
\tag{A28}
$$

which we use as the proposal distribution in the MH algorithm. We obtain $\tilde{h}^*_{1:T+1}$ in (A28) with the same search algorithm as in the case of the NCP form and apply the block sampler to improve the acceptance rate in the MH algorithm.

Appendix A.2.2. Regression Coefficients $\beta$

By ignoring the terms that do not depend on $\beta$, we can rearrange the density of $Y_t$ in (15) as

$$
p(y_t|\tilde{h}_t, \tilde{h}_{t+1}, \beta, \theta_{-\beta}) \propto \exp\left[-\frac{1}{2}(\tilde{y}_t - \tilde{x}'_t\beta)^2\right],
$$

where

$$
\tilde{y}_t = y_t \exp(-\tilde{h}_t) - \gamma(\tilde{h}_{t+1} - \phi\tilde{h}_t), \quad \tilde{x}_t = -\gamma(x_{t+1} - \phi x_t).
$$

By defining $\tilde{y} = [\tilde{y}_1; \ldots; \tilde{y}_T]$ and $\tilde{X} = [\tilde{x}'_1; \ldots; \tilde{x}_T]$, we have

$$
\begin{aligned}
p(y_{1:T}|\tilde{h}_{1:T+1}, \beta, \theta_{-\beta}) &= \prod_{t=1}^{T} p(y_t|\tilde{h}_t, \tilde{h}_{t+1}, \beta, \theta_{-\beta}) \\
&\propto \exp\left[-\frac{1}{2}(\tilde{y} - \tilde{X}\beta)'(\tilde{y} - \tilde{X}\beta)\right].
\end{aligned}
\tag{A29}
$$

Then the conditional posterior distribution of $\beta$ is given by

$$
\begin{aligned}
& p(\beta|\tilde{h}_{1:T+1}, \theta_{-\beta}, y_{1:T}) \\
& \propto p(y_{1:T}|\tilde{h}_{1:T+1}, \beta, \theta_{-\beta})p(\tilde{h}_{1:T+1}|\beta, \theta_{-\beta})p(\beta) \\
& \propto \exp\left[-\frac{1}{2}(\tilde{y} - \tilde{X}\beta)'(\tilde{y} - \tilde{X}\beta) - \frac{1}{2\tau^2}(\tilde{h}_{1:T+1} - X\beta)'V(\tilde{h}_{1:T+1} - X\beta)\right. \\
& \left. \qquad\quad - \frac{1}{2}(\beta - \bar{\mu}_\beta)'\bar{\Omega}_\beta(\beta - \bar{\mu}_\beta)\right].
\end{aligned}
\tag{A30}
$$

By completing the square, we have the conditional posterior distribution of $\beta$ as

$$\beta \sim \text{Normal}\left(\tilde{\mu}_\beta, \tilde{\Sigma}_\beta\right), \tag{A31}$$

where

$$\tilde{\Sigma}_\beta = \left(\tilde{X}'\tilde{X} + \frac{1}{\tau^2}X'VX + \bar{\Omega}_\beta\right)^{-1},$$

$$\tilde{\mu}_\beta = \tilde{\Sigma}_\beta\left(\tilde{X}'\tilde{y} + \frac{1}{\tau^2}X'V\tilde{h}_{1:T+1} + \bar{\Omega}_\beta\bar{\mu}_\beta\right).$$

Since $\tilde{X}'\tilde{X} = \gamma^2 X'VX$,

$$\tilde{\Sigma}_\beta = \left(\left(\gamma^2 + \frac{1}{\tau^2}\right)X'VX + \bar{\Omega}_\beta\right)^{-1},$$

$$\tilde{\mu}_\beta = \tilde{\Sigma}_\beta\left(\tilde{X}'\tilde{\epsilon} + \left(\gamma^2 + \frac{1}{\tau^2}\right)X'V\tilde{h}_{1:T+1} + \bar{\Omega}_\beta\bar{\mu}_\beta\right),$$

where $\tilde{\epsilon} = [\tilde{\epsilon}_1; \ldots; \tilde{\epsilon}_T]$.

Appendix A.2.3. Leverage Parameter $\gamma$

Replacing $\epsilon_t$ and $\eta_t$ in (19) with $\tilde{\epsilon}_t$ and $\tilde{\eta}_t$, respectively, we have

$$\gamma|\tilde{h}_{1:T+1}, \theta_{-\gamma}, y_{1:T} \sim \text{Normal}\left(\frac{\sum_{t=1}^T \tilde{\eta}_t\tilde{\epsilon}_t + \bar{\omega}_\gamma\bar{\mu}_\gamma}{\sum_{t=1}^T \tilde{\eta}_t^2 + \bar{\omega}_\gamma}, \frac{1}{\sum_{t=1}^T \tilde{\eta}_t^2 + \bar{\omega}_\gamma}\right). \tag{A32}$$

Appendix A.2.4. Variance $\tau^2$

It is straightforward to show that the conditional posterior distribution of $\tau^2$ is

$$\tau^2|\tilde{h}_{1:T+1}, \theta_{-\tau^2}, y_{1:T} \sim \text{Inv. Gamma}\left(\frac{T+1}{2}a_\tau, \frac{1}{2}(\tilde{h}_{1:T+1} - X\beta)'V(\tilde{h}_{1:T+1} - X\beta) + b_\tau\right). \tag{A33}$$

Appendix A.2.5. AR(1) Coefficient $\phi$

Replacing $h_t$ in derivation of (A21) with $\tilde{h}_t - x_t'\beta$, we have

$$\phi \sim \text{Normal}\left(\frac{\sum_{t=1}^T (\tilde{h}_{t+1} - x_{t+1}'\beta)(\tilde{h}_t - x_t'\beta)}{\sum_{t=2}^T (\tilde{h}_t - x_t'\beta)^2}, \frac{\tau^2}{\sum_{t=2}^T (\tilde{h}_t - x_t'\beta)^2}\,\middle|\, -1 < \phi < 1\right). \tag{A34}$$

We use (A34) as the proposal distribution for $\phi$ in the MH algorithm.

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
