# Peer review of "Bayesian Analysis of Intraday Stochastic Volatility Models of High-Frequency Stock Returns with Skew Heavy-Tailed Errors"

_jrfm, doi:10.3390/jrfm14040145_

Round 1

Reviewer 1 Report

see the pdf file attached. 

Author Response

Thank you for your comments and suggestions.   Since the application of Bernstein polynomials and ASIS are the best selling points of this study, we are honored to see your comment. At the same time, we reflected on the insufficient discussion of the previous studies. We added more references and discussions about the related literature and strengthened the manuscript.   1. These three early studies on SV are very important. We added their citations. 2. A sequential Monte Carlo used in Virbickait et al. (2019) is a good method as an alternative approach. We added citation and discussion about it to the manuscript. 3. As you pointed out, realized volatility (RV) should be described for comparison. We added a discussion about the RV model and mentioned some references. 4. We are sorry about incorrect references. We checked and revised the manuscript again.

Reviewer 2 Report

This manuscript presents a high-frequency Bayesian analysis using the stochastic volatility model with skew heavy-tailed innovations. Overall, the paper is well designed and written. I hope my following comments may help the authors to further clarify some points and revise their research.

  1. The current paper is very long. Sections 2 and 3 may contain excessive information than needed. The authors might want to move some secondary descriptions to the appendices, and only keep the most relevant contents of the modelling and estimation processes.
  2. The font sizes of the tables and figures are too small. Those should be enlarged to a similar size to that of the main texts. Doing so will improve the readability of the paper. Also, the styles of the tables should be consistent. I see, for example, toprules are used in some, but skipped in others.
  3. I understand that the authors want to use the data to show the importance of the skewed heavy-tail errors. However, the motivation of suspecting that the Brexit will impact the Japanese market is a bit weak. It might be better to assess the influence on the FTSE100 (for example), which is much more relevant to the event of Brexit. Also, with the existence of such a major event, a regime-switching (or other structural-breaks) model could be more appropriate. I do not expect the authors to provide more empirical results on this comment, but more clarifications/motivations would improve the comprehensiveness of this paper.
  4. A thorough English check on the spelling and other grammatical errors should be performed. Also, there are some incorrect references. For instance, [7] has no authors' information.

Author Response

Thank you for your comments and suggestions.   1. We moved detailed descriptions of the Markov chain sampling algorithm to the appendices and keep the most relevant contents in the main text. 2. We increased the font sizes of the figures and tables. Also, we unified the styles of the tables. 3. As you pointed out, the importance of using data of the Japanese market should be more emphasized. The reason why we use the data of the Tokyo Stock Exchange (TSE) is that it is the only market among three major stock markets (Tokyo, London and New York) which opened when the result of the Brexit referendum was announced during the trading hours. We added a few sentences that explained the importance of analyzing the TSE to Section 4. We also agree that a regime-switching model is a promising approach to be considered. In this study, however, we regard the change in volatility due to the announce of the Brexit referendum not as a structural change but as a temporary shock. We could apply a regime-switching model as our future works. 4. We checked grammar and spelling again. Also, we are sorry about incorrect references. We checked and revised the manuscript.

Reviewer 3 Report

It was a pleasure reading the paper and hope more papers on this quality will be submitted to JRFM.

A comment on the current version: The normalization of the log-return time series to have zero mean and variance 1 is strange, to say-the-list. First, how the normalization is done? By subtracting the  sample mean and dividing by the standard deviation? But with the suggested  stochastic model for log- returns, the observed sample of log-returns is far from  iid observations. Also, the suggested model exhibits volatility clustering, leverage  effect, seasonality and heavy tails. Does then the sample mean and sample variance even weakly converge  to the theoretical mean and variance? And within the sample size how close the sample mean and the variance are  to the theoretical ones?

I am not blaming the authors for that omission  as such "normalization" is often used in the academic literature, but  makes zero sense.  Why am I saying "zero sense"?  I do trade intraday ( that is, I care about the "alpha") , and over 30 years I was (and am) closely involved in commercial financial risk management products ( that is, I do care about the "beta"). None of our commercial clients would like to see the log-returns series "normalized". By subtracting the "sample mean" (whatever that means), one kills the "alpha". By dividing by the "sample standard deviation" ( again, whatever that means) one kills the valuable information of the time-varying volatility, and thus, kills the "beta". To tell that one can then go back to the original theoretical model by inverting the Z-transformation is again nonsense, by the same arguments, I have already pointed out above. The reason that no risk manager, would study the "normalized" log-return series  is because she wants to see the "clean" volatility estimates, and evaluate option contracts  (puts, futures) that will mitigate the risk.

The last paragraph leads to my suggestion for hedging the estimated tail-risk. Again, as practitioner, seeing a good estimate of  the tail risk ( the one I mainly care) , that  is, the estimates of  1 minute VaR(99%) and CVaR(99%), is just 20% of my job as CRO. I have to report to the CIO how to perform risk budgeting and do hedging of the tail risk, and those two tasks,  should be done  WITHIN THE SAME STOCHASTIC MODEL I HAVE ESTIMATED THE TAIL RISK . This is a must for any financial risk vendor, if she tries to approach HFs like Citadel

https://www.citadel.com/

https://www.youtube.com/watch?v=cNWshKyfHBg

with her intraday risk management system. My two cents advice to the authors: If there is no follow-up of your paper with a risk budgeting and hedging within the stochastic model you have backtested ( again, within the framework of your model) ,  your paper will be one of the pile of others on financial risk assessment, which practitioners do not have to even look at.

There are standard approaching to option pricing in a setting similar to that in the paper, see  for example:

1.  https://hal.archives-ouvertes.fr/hal-00511965/document, 

2. https://papers.ssrn.com/sol3/papers.cfm?abstract_id=946002

3. https://academic.oup.com/jfec/article-abstract/16/3/425/4555529

4.https://www.researchgate.net/publication/304066698_

Hope my comments are helpful.

Author Response

Thank you for your comments and suggestions.   We agree that "normalization" should not be used when we are concerned with alpha, beta, etc. The reason we use normalization in our study is to optimize the efficiency of numerical calculations. Since non-normalized results can be reproduced by multiplying the standard deviation and adding the mean to the normalized ones, we think our approach is still defensible. The suggestions from your experience as a practitioner is very valuable for us to consider our future research. There is no doubt that the tail risk is one of the biggest concerns in both financial study and practice. Your advice that our modeling can be applicable to option pricing is also helpful.   With practicality in mind, we will tackle with risk analysis and option pricing as our future challenges.

Round 2

Reviewer 1 Report

Good revision.